# A signal cascade originated from epidermis defines apical-basal patterning of Arabidopsis shoot apical meristems

Han Han[1,2,7], An Yan[3,4,7], Lihong Li[2,5], Yingfang Zhu[6], Bill Feng[3], Xing Liu[2,5] & Yun Zhou [1,2✉]

In multicellular organisms, a long-standing question is how spatial patterns of distinct cell types are initiated and maintained during continuous cell division and proliferation. Along the vertical axis of plant shoot apical meristems (SAMs), stem cells are located at the top while cells specifying the stem cells are located more basally, forming a robust apical-basal pattern. We previously found that in *Arabidopsis* SAMs, the HAIRY MERISTEM (HAM) family transcription factors form a concentration gradient from the epidermis to the interior cell layers, and this gradient is essential for the stem cell specification and the apical-basal patterning of the SAMs. Here, we uncover that epidermis specific transcription factors, ARABIDOPSIS THALIANA MERISTEM LAYER 1 (ATML1) and its close homolog, define the concentration gradient of HAM in the SAM through activating a group of microRNAs. This study provides a molecular framework linking the epidermis-derived signal to the stem cell homeostasis in plants.

[1] Department of Botany and Plant Pathology, Purdue University, West Lafayette, IN 47907, USA. [2] Purdue Center for Plant Biology, Purdue University, West Lafayette, IN 47907, USA. [3] Division of Biology and Biological Engineering, California Institute of Technology, Pasadena, CA 91125, USA. [4] Howard Hughes Medical Institute, California Institute of Technology, Pasadena, CA 91125, USA. [5] Department of Biochemistry, Purdue University, West Lafayette, IN 47907, USA. [6] Institute of Plant Stress Biology, State Key Laboratory of Cotton Biology, Department of Biology, Henan University, Kaifeng, China. [7] These authors contributed equally: Han Han, An Yan. ✉email: zhouyun@purdue.edu

Plant shoot apical meristems (SAMs) is the sustainable resource for the shoot and flower development[1]. In the model plant Arabidopsis, several key pathways in control of stem cell homeostasis in the SAMs have been identified[2–4]. Among them, the HAIRY MERISTEM (HAM) family GRAS (GAI, RGA and SCR) domain transcription factors play essential roles in determining the specification and proliferation of stem cells in the SAMs[4–9]. Two HAM family members, HAM1 and HAM2 proteins, interact with the homeodomain transcription factor WUSCHEL (WUS) and function together with WUS to regulate downstream gene expression and maintain the stem cell homeostasis[7,8]. In addition, HAM1 and HAM2 keep the CLV3 expression off at the inner cell layers to establish an apical-basal polarity of the SAMs and the de novo axillary stem cell niches[9,10]. Through both computational modeling and experimentation, we previously found that HAM1/2 are expressed with a concentration gradient from the epidermis to the deep cell layers in the SAMs, which is important for the stem cell homeostasis in the established SAM[9]. In the de novo formation of axillary meristems, this HAM gradient is formed from early to late developmental stages and it is essential for the patterning of newly initiated stem cell niches. However, how this concentration gradient of HAM is defined in the developing meristem and continually maintained in the established SAMs remains to be elucidated.

The conserved microRNA171 (miR171) family in Arabidopsis has been reported to specifically repress the HAM family genes[11–13]. miR171, the 21-ribonuleotide species, specifically recognizes and cleaves HAM1, HAM2, and HAM3[11,12]. miR171 originates from the products of four genes in Arabidopsis including MIR171A, MIR171B, MIR171C, and MIR170[14–16]. Several studies have reported that miR171 is expressed in a wide range of tissues including cotyledons, hypocotyls, leaves, meristems, and flowers[11,12,17,18]. One recent study reported that MIR171A, one of MIR171 genes, is expressed in the epidermal layer during the embryogenesis and vegetative development stages[19]. In this work, we establish a molecular linkage between the epidermis specific transcription factors, including ARABIDOPSIS THALIANA MERISTEM LAYER 1 (ATML1) and its close homolog PROTODERMAL FACTOR 2 (PDF2)[20–28], and the expression of MIR171. Combining in vitro biochemistry, in vivo live imaging, and in silico modeling approaches, we uncover the mechanism underlying how the epidermis expressed ATML1/PDF2 patterns the concentration gradient of HAM via miR171 in the established SAMs and de novo initiating stem cell niches.

## Results

### MIR171/170 are specifically expressed in the SAM epidermis.
To define and compare the expression patterns of three MIR171 genes and one MIR170 gene in the SAMs, we generated transcriptional reporters for these genes in which a fluorescent protein is placed in between their own 5′ promoters and 3′ terminators (Fig. 1a–d). Through the confocal live imaging of the SAMs using identical settings, surprisingly, we found that in the SAMs and the floral meristems (FMs), the expressions of all four reporters are restricted to the epidermis. In the SAMs, MIR171A and MIR171B (pMIR171A::H2B-GFP and pMIR171B::H2B-GFP) are expressed at high levels (Fig. 1a (panels 1–3), Fig. 1b (panels 1–3)) (Supplementary Movies 1 and 2), while MIR171C and MIR170 (pMIR171C::H2B-GFP and pMIR170::H2B-GFP) are expressed at very low levels (Fig. 1c (panels 1–3), Fig. 1d (panels 1–3)) (Supplementary Movies 3 and 4). In the young seedlings, pMIR171A::H2B-GFP and pMIR171B::H2B-GFP are also specifically expressed in the epidermis of the developing leaves and hypocotyls (Fig. 1a (panels 4–5), Fig. 1b (panels 4–5). In contrast,

the pMIR171C::H2B-GFP reporter activity is undetectable from young leaves and hypocotyl in the seedling, and pMIR170::H2B-GFP is weakly expressed in the developing young leaves and hypocotyls (Fig. 1c (panels 4–5), Fig. 1d (panels 4–5)). These results showed that similar to MIR171A[19], MIR171B expression is strongly activated and specifically confined in the epidermal layer, whereas, the expression of MIR171C and MIR170 is also specifically restricted at epidermis of the SAM but their expression level is very low (Fig. 1a–d).

In addition, consistent with our previous report[9], the HAM2 translational reporter pHAM2::YFP-HAM2 in ham123 showed a concentration gradient in the SAM (Fig. 1e) (Supplementary Movie 5). However, the HAM2 transcriptional reporter pHAM2::H2B-GFP, which lacks the miR171 binding site, is highly expressed in all cell layers of the SAM (Fig. 1f) (Supplementary Movie 6). These results suggest that miR171 keeps HAM off in the L1 layer and it further shapes the HAM gradient, likely through the movement from L1 to L2 or even upper layers of the corpus to suppress HAM.

### Epidermis specific ATML1/PDF2 directly upregulate MIR171A/B.
The expression patterns of the MIR171A and MIR171B reporters in epidermis of the SAMs (Fig. 1a, b) are almost identical to that of the ATML1 and PDF2[20,21], which are epidermis specific transcription factors. Thus, we decided to examine whether these two genes are directly regulated by ATML1 and PDF2.

First, we tested the interaction between ATML1 or PDF2 protein and the promoter DNA of MIR171A using the yeast one hybrid (Y1H) system (Fig. 2a, b). Both the β-galactosidase liquid and x-gal lifting assays showed that ATML1 and PDF2 proteins interact with the full-length promoter of MIR171A (Fig. 2a, b, Supplementary Fig. 1a). Because ATML1/PDF2 was reported to bind to an L1 box motif[22,29], we searched along the MIR171A promoter for putative ATML1/PDF 2 binding sites. We used the conserved DNA sequences from the previously reported L1-box motifs[22,29] to perform the search and we identified three regions that contain the conserved TT(A/T)AATG(C/T) sequences (Fig. 2a). We then generated new reporters for Y1H assays, each of which contains a fragment of the MIR171A promoter DNA that includes one putative L1 box (Fig. 2a, c–e). The Y1H assays showed that all three fragments (F1, F2 and F3) interact with both ATML1 and PDF2 (Fig. 2c–e). We further confirmed these protein-DNA interactions by the electrophoresis mobility shift assays (EMSA). We labeled these DNA fragments with Cy5 (Fig. 2a), and when each of the Cy5-labeled probe was incubated with recombinant $^{GST}$ATML1$^{30–134}$, the DNA binding domain of the ATML1 (Supplementary Fig. 2), the Cy5-labeled DNA probe with reduced gel mobility representing the protein-DNA complex was observed in the native gel (Fig. 2f–k). This shifted band was not observed when GST was incubated with the Cy5-labeled probe (Fig. 2f, h, j), nor when excess amount of additional unlabeled probe (competitor) was supplied prior to the incubation with $^{GST}$ATML1$^{30–134}$ (Fig. 2g, i, k), demonstrating the specificity of the interaction in EMSA. We further examined whether MIR171A can be directly regulated by ATML1 using a dual-luciferase reporter assay. Compared to empty vector controls, the expression of the pMIR171A::LUC reporter is significantly induced by the ATML1 effector in the tobacco transient expression system (Fig. 2l).

Y1H also showed that ATML1 and PDF2 proteins interact with the full-length promoter of MIR171B (Fig. 3a, b, Supplementary Fig. 1b). We next found five putative binding sites of ATML1/PDF2 on the promoter sequence of the MIR171B gene, from either forward or reverse directions (Fig. 3a), which all share at least six conserved base pairs of T (A/T)AATG with the

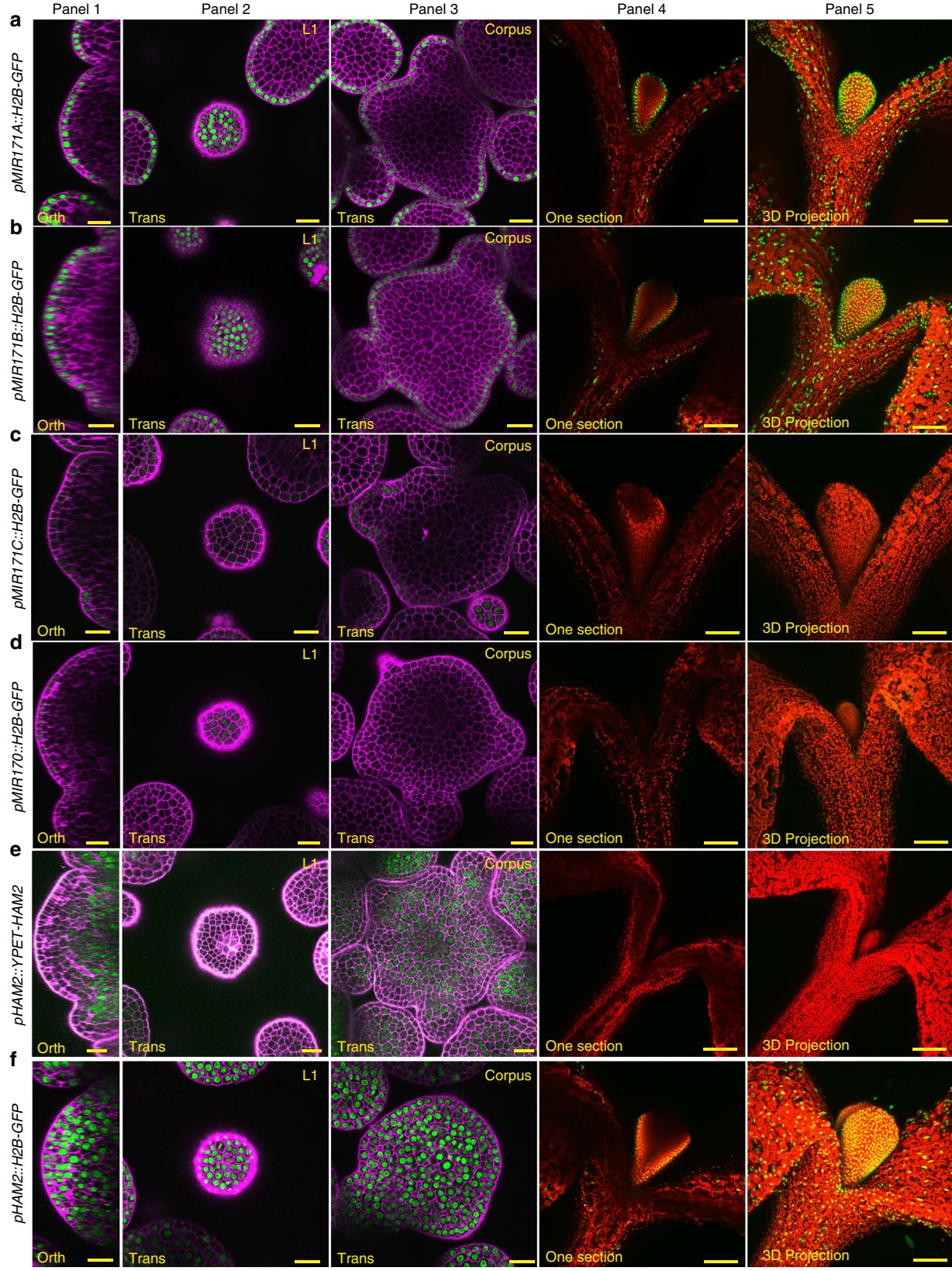

previously identified L1 box[22,29]. We tested the binding of ATML1/PDF2 with each DNA fragment (from F1 to F5) that contains a putative binding site (Fig. 3a, c, d), and we found that two fragments (F3 and F4) interact with ATML1 and PDF2 in

Y1H (Fig. 3c, d). We further confirmed the interaction between these two fragments and the recombinant $^{GST}$ATML1$^{30-134}$ protein using EMSA, with both the GST and unlabeled competitor probes as the controls (Fig. 3e–h). All the results

**Fig. 1 Epidermis expressed *MIR170/171* genes define the apical-basal gradient of HAM in *Arabidopsis* SAMs. a** The *pMIR171::H2B-GFP* reporter is highly expressed at the epidermis in the SAM (panels 1–3) and in the young leaves (panels 4–5). **b** The *pMIR171B::H2B-GFP* reporter is highly expressed at the epidermis in the SAM (panels 1–3) and in the young leaves (panels 4–5). **c** The *pMIR171C::H2B-GFP* reporter is weakly expressed at the epidermis in the SAM (panels 1–3) and undetectable in the young leaves (panels 4–5). **d** The *pMIR170::H2B-GFP* reporter is weakly expressed at the epidermis in the SAM (panels 1–3) and in the young leaves (panels 4–5). **e** The *pHAM2::YPET-HAM2* reporter that is sensitive to the miR170/171 is expressed with a concentration gradient in the SAM, with high expression at inner cell layers and no expression in epidermis (panels 1–3). The *pHAM2::YPET-HAM2* reporter is not expressed at the epidermis of the young leaves (panels 4–5). **f** The *pHAM2::H2B-GFP* reporter that is not sensitive to miR170/171 is highly expressed in all the cell layers (including the epidermal layer) in the SAM (panels 1–3), and the *pHAM2::H2B-GFP* reporter is highly expressed at the epidermis of the young leaves (panels 4–5). Panels (from left to right): 1, orthogonal view of a SAM; 2, transverse section view of L1 in the SAM; 3, transverse section view of corpus in the SAM; 4, section view of a seedling; 5, 3D projection view of the seedling. Channels: GFP/YFP (green), propidium iodide (PI) counterstain (purple), and Chlorophyll (red). Scale bar: (panels 1–3) 20 μm; (panels 4–5) 100 μm.

consistently demonstrated that F3 and F4 with the core T (A/T) AATG sequence can interact with ATML1 but other fragments containing similar sequences cannot, suggesting that the flanking sequences of the T (A/T)AATG motif also contribute to the interaction between ATML1 and the DNA. In the dual-luciferase transactivation assay, we found that ATML1 can activate the *pMIR171B::LUC* reporter (Fig. 3i), and the fold change of activation over the empty vector control (Fig. 3i) is comparable to that in the assay with a *pMIR171B::LUC::3′MIR171B* reporter (Supplementary Fig. 3). These results suggest that the 3′ regulatory region of *MIR171B* gene is dispensable for the activation of *MIR171B* by ATML1, although an additional L1 box from this region also interacts with both ATML1 and PDF2 (Supplementary Fig. 4). Taken together, our results demonstrated that ATML1 activates *MIR171A* and *MIR171B* through directly binding to their promoters.

Besides *MIR171A* and *MIR171B*, ATML1 also binds to the promoters of *MIR171C* (Fig. 4a–c) and *MIR170* in the Y1H assays (Fig. 4d–f). We found that one fragment (F1) from the *MIR171C* promoter (Fig. 4a) and one fragment (F2) from the *MIR170* promoter (Fig. 4d) contain the T(A/T)AATG(C/T) sequence. Both fragments showed strong interactions with ATML1 in Y1H (Fig. 4c, f). These results may explain why the expression of the *MIR171C* and *MIR170* reporters is also specific in the epidermis of SAMs, though their expression levels are low (Fig. 1c, d).

**L1-box and ATML1/PDF2 are essential for regulating *MIR171*.** To investigate whether these putative L1 box elements in the *MIR171A* promoter are essential for the epidermis specific expression of miR171, we mutated two L1-boxes in the *MIR171A* promoter and generated a *pMIR171A-mut* reporter (Supplementary Fig. 5). In the Y1H assay, these mutations significantly reduced the interaction between the *pMIR171A* promoter DNA and the ATML1 (Fig. 5a) or PDF2 protein (Fig. 5b). Furthermore, the mutations greatly reduced the activation of the *pMIR171A-mut::LUC* reporter by ATML1 in the transient dual-luciferase assays (Fig. 5c). To further examine the biological significance of these L1 box elements in the *MIR171A* promoter, we introduced the *pMIR171A-mut::H2B-GFP* reporter in *Arabidopsis* (Supplementary Fig. 5). Compared to the expression of the *pMIR171A::H2B-GFP* control (Fig. 5d–g), the *pMIR171A-mut::H2B-GFP* reporters showed reduced expression (Fig. 5h–k, Supplementary Fig. 6). Specifically, the majority of the independent transgenic lines we obtained (26 out of 31, 83.9%) showed no expression of the *pMIR171A-mut::H2B-GFP* reporter in SAMs but only weak expression in the epidermis of sepal primordia (Fig. 5h–k, Supplementary Fig. 6). Thus, we conclude that the L1 boxes mediate the activation of *MIR171A* by ATML1 and PDF2 in the epidermis of the SAMs and FMs.

To investigate whether ATML1 and PDF2 are required for the epidermis specific expression of the *MIR171A* and *MIR171B*, we introduced the *pMIR171A::H2B-GFP* reporter or the *pMIR171B::H2B-GFP* reporter into the *atml1-1 pdf2-1* double mutant[23] through genetic crosses. Compared to the expression of *pMIR171A::H2B-GFP* and *pMIR171B::H2B-GFP* in wild type plants (Fig. 6a–e, k–o), the expression levels of these two reporters were greatly reduced in *atml1-1 pdf2-1* double mutant seedlings (Fig. 6f–j, p–t). These results demonstrated that *ATML1* and *PDF2* are essential for the expression of *MIR171A* and *MIR171B*.

**3D computational model for the L1-miR171-HAM signal cascade.** Based on our results that ATML1/PDF2—the L1 specific transcription factors—directly activate miR171 transcription (Figs. 1–6) and the previous findings that miR171 directly silences HAM[11,12], we hypothesized that there exists a L1-miR171-HAM signaling cascade, in which the epidermal ATML1/PDF2 determine the apical-basal concentration gradient of HAM in the SAM through the regulation of *MIR171A/B*. To test this hypothesis, we developed a computational model, which derives from our previously reported SAM model[9], to help understand this signaling cascade in different cell layers in the three dimensional (3D) SAM. Different from the previous model, in which we used HAM concentration gradient as a key functional input[9], we now used the epidermal specification factors, ATML1 and PDF2, as the key input. As an output, we simulated the concentration gradient of HAM in the SAM, as well as the *MIR171* promoter activity and the patterns of miR171 (Supplementary Fig. 7).

For simplicity of computational model, we first defined the functional ATML1 and its homolog PDF2 proteins as the input (ML1p), which are well-known to be L1-layer specific[20,21]. In the SAM of the wild-type plant, we set the concentration of ML1p 1 a.u. in cells of the L1 layer, and 0 a.u. in cells of the other layers.

To model the dynamics of miR171, we set three conditions. First, the production of miR171 RNA is activated by the L1 layer specific ATML1 protein. As a result, miR171 is not produced in cells at the L2 and corpus layers in a wild type SAM. This condition is consistent with our experimental results described above (Fig. 1a–d). Second, we set a constant degradation rate for miR171. Third, we set a low rescaled (by cell size) diffusion constant for the movement of miR171 from epidermis to deep layers. This condition is consistent with results from our RNA in situ hybridization experiments, which showed that the level of the endogenous miR171 is high in L1 and low in L2 and upper corpus layers (Supplementary Figs. 8 and 9).

For the dynamics of HAM mRNA, we also set three conditions. First, the transcription of HAM mRNA is constitutively active in all layers of SAM. This is based on our finding that the transcriptional reporter of *pHAM2::H2B-GFP* lacking the miR171 binding site is highly expressed in all cell layers of the SAM

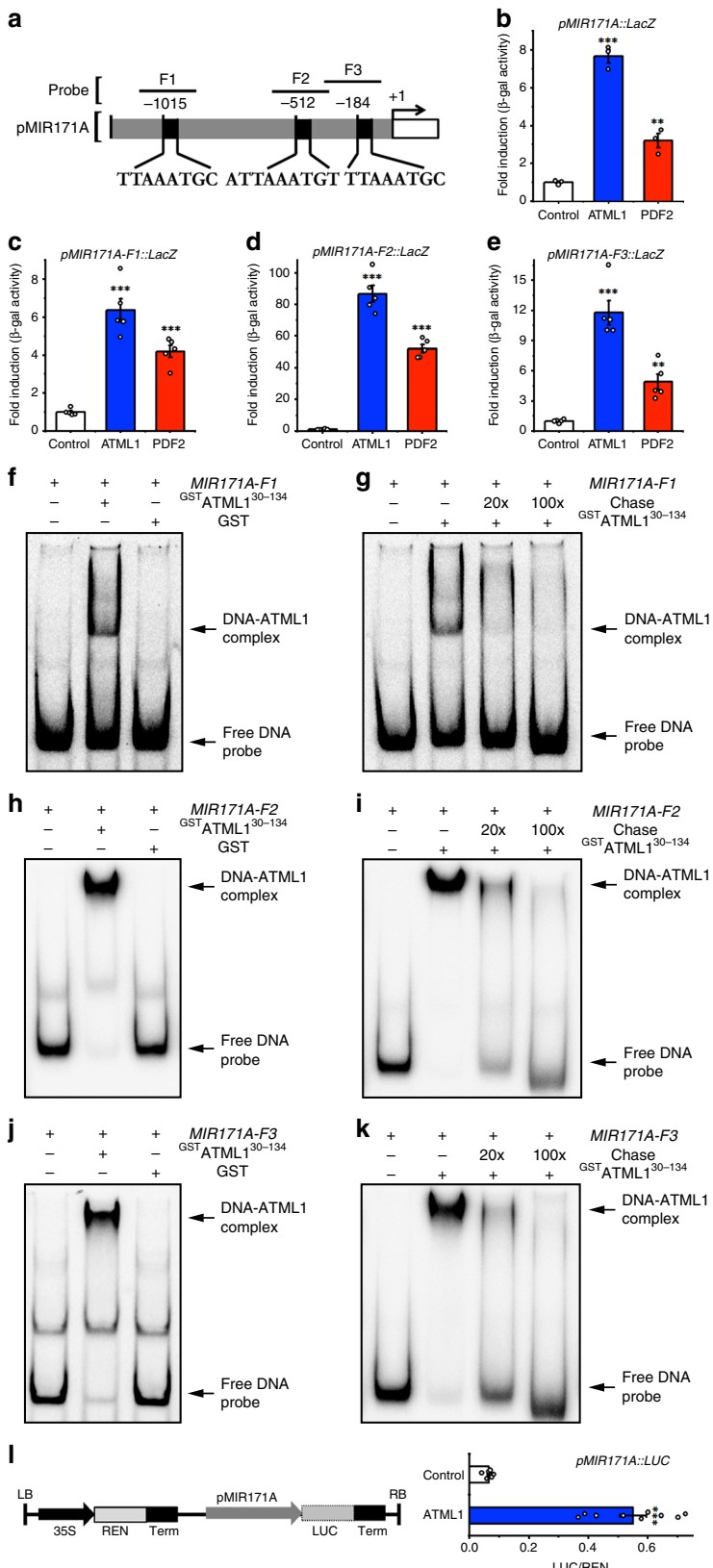

(Fig. 1f). Second, we set a constant rate for the miR171 independent *HAM* mRNA degradation. Third, miR171 activates the degradation of *HAM* mRNA, which is well supported by previous reports on the function of miR171[11–13].

For the dynamics of HAM protein, we defined that the synthesis rate of HAM protein depends on the level of *HAM*

mRNA, and the HAM protein is degraded at a constant degradation rate.

In addition to miR171, *HAM* mRNA and HAM protein, the three key components, we also included an H2B-GFP transcriptional reporter to reflect the activity of the *MIR171* promoter. Similar to miR171, the transcription of the *H2B-GFP* mRNA is

**Fig. 2 ATML1 and PDF2 directly activate *MIR171A*. a** Schematic structure of the *MIR171A* promoter. The expanded diagram shows the sequences of putative L1 boxes on the promoter. Probes indicate the DNA fragments used in the EMSA and the yeast one hybrid (Y1H) assay, and +1 indicates the transcription start site. **b** The full-length promoter of *MIR171A* interacts with ATML1 and PDF2 proteins in Y1H assays. Y-axis: the relative activity of the *pMIR171A:: lacZ* reporter. All the numbers are normalized to the average value of the empty vector control. Bars: mean ± standard error (SE) ($n = 3$ biological replicates). **$P < 0.01$, ***$P < 0.001$ (Student's two-tailed $t$-test). **c**–**e** Different DNA fragments from the *MIR171A* promoter interact with ATML1 and PDF2 proteins in Y1H. Y-axis: the relative activity of the *pMIR171A-F1::lacZ* (**c**), *pMIR171A-F2::lacZ* (**d**) and *pMIR171A-F3:: lacZ* (**e**) reporters. All the numbers are normalized to the average value of the empty vector control. Bars: mean ± SE ($n = 5$ biological replicates). **$P < 0.01$, ***$P < 0.001$ (Student's two-tailed $t$-test). **f**–**k** Electrophoretic mobility shift assay (EMSA) shows that $^{GST}ATML1^{30-134}$ but not GST binds to the Cy5-labeled probes including *MIR171A-F1* (**f**), *MIR171A-F2* (**h**) and *MIR171A-F3* (**j**). The $^{GST}ATML1^{30-134}$–induced mobility shift of the Cy5-labeled DNA probe can be chased away when excess amount of the unlabeled DNA probe (chase) is present (**g**, **i**, **k**). Arrows indicate free DNA probes and DNA-protein complexes. **l** Dual-luciferase assays show that ATML1 activates the transcription of *MIR171A*. Left panel: structure of the reporter construct. Term, CaMV terminator; LB, transfer DNA (T-DNA) left border; RB, T-DNA right border. Right panel: LUC/REN, ratio of firefly luciferase (LUC) to Renilla luciferase (REN) activity. The LUC/REN in tobacco cells co-transformed with the reporter *pMIR171A::LUC* and the ATML1 effector ($n = 9$ biological replicates) is significantly higher than that in the cells co-transformed with the same reporter and empty vector control ($n = 7$ biological replicates). Bar: mean ± SE. ***$P < 0.001$ (Student's two-tailed $t$-test). Source data underlying Fig. 2b–l are provided as a Source Data file.

activated by ATML1 protein, and the *H2B-GFP* mRNA is degraded at a constant rate. Further, the synthesis rate of the H2B-GFP reporter protein depends on the level of its mRNA, and it is degraded at a constant rate. Unlike miR171, which moves into deeper layers of the SAM, the H2B-GFP reporter cannot move between cells.

Based on the model conditions mentioned above, we established five dynamic equations for miR171, *HAM* mRNA, HAM protein, *H2B-GFP* mRNA, and H2B-GFP protein, respectively (see details in Methods). In this model, the miR171 and *HAM* mRNA are essential for the apical-basal pattern of *HAM* gene expression. Next, we set out to test whether the model can simulate the pattern of *HAM* mRNA in a wild-type SAM using one set of parameters (Supplementary Table 1), and we found that our model reproduced the typical apical (low)-basal (high) gradient of the *HAM* mRNA (Fig. 7m).

Our model contains six key parameters that control the dynamics of miR171 and the *HAM* mRNA expression patterns (Supplementary Table 2). We then explored the working ranges of these six key parameters surrounding the defined parameter values (Supplementary Table 2). In total, we found 235 sets of different combinations of parameter values (Supplementary Data 1). Using each set of parameter values, we can simulate patterns of *HAM* mRNA that are qualitatively comparable to the observed pattern in a wild-type SAM (Supplementary Movies 7–12), demonstrating the system we established is robust. These 235 sets of parameter values were also used in the sensitivity analyses for the six key parameters (Supplementary Fig. 10). We found that the total amount of *HAM* mRNAs shows higher sensitivity to changes in three of the six parameters, including $k_{hrp}$ and $k_{hrnh}$ that directly control the production and degradation of *HAM* mRNA, respectively, and $k_{mirn}$ that directly determines the degradation of miR171 (Supplementary Fig. 10).

In addition to the local parameter search described above (Supplementary Table 2, Supplementary Data 1), we further carried out unbiased random search for more diverse parameters in a 100-fold range (See Methods). We randomly sampled 20,000 sets of six different parameters uniformly on the log-scale (Supplementary Data 2) and ran simulations using each of 20,000 random parameter sets in the 3D template. We found that 173 new sets of solutions (Supplementary Data 3), which are different from the initial set of parameters (Supplementary Table 1), all lead to *HAM* mRNA expression patterns qualitatively comparable to the experimental observation in a wild type SAM (Supplementary Movie 13). We performed the sensitivity analyses for the six parameters again with the 173 new sets of parameter values, and the results (Supplementary Fig. 11) show the same tendencies for

the parameter sensitivities compared to the analyses mentioned above (Supplementary Fig. 10).

**Evaluating the L1-miR171-HAM cascade in silico and in vivo.** To further study the regulatory cascade in the SAM, we computationally predicted the patterns of miR171 and *HAM* gene expression when the ATML1 protein is ectopically activated in the SAM. We first defined one set of [ML1p] input in the model based on the experimental results from the *ATML1* RNA in situ hybridization (Supplementary Fig. 12). When ATML1 is ectopically activated, we set the level of ML1p 1.1 a.u. in the epidermal layer and 0.4 a.u. in sub-epidermal and corpus layers (see Methods for details), and we kept the values of all parameters the same for the simulation (Supplementary Table 1). As an output, the model shows that *MIR171* promoter activity is increased in all layers of SAM while *HAM* mRNA levels are dramatically decreased in the SAM (Fig. 7a–d, m, n). We then explored more ML1p input patterns and we found that 75 different ML1p input patterns all lead to dramatic reduction of *HAM* mRNA levels in all layers of SAM (Supplementary Movies 14 and 15), which is insensitive to differences of [ML1p] between epidermis and deeper layers, suggesting a robust response in the model for the ectopic activation of ATML1.

In parallel, we tested these computational predictions experimentally (Fig. 7e–h, Supplementary Figs. 12–15). Because ATML1 drives the epidermal specification pathway and it regulates different sets of downstream targets and developmental processes[22,23,25–28], prolonged activation of ATML1 will likely result in perturbations in a wide range of developmental events. Therefore, we decided to apply a dexamethasone (Dex) inducible transient activation system (*ATML1-GR*) to examine the immediate effects of ATML1 activation on the expression of *MIR171* and *HAM*. We first introduced the *35S::ATML1-GR* expression cassette in the *pMIR171B::H2B-GFP* reporter line, and then we performed the time-lapse live imaging of the *pMIR171B:: H2B-GFP* reporter in the SAMs with either the mock or Dex treatment over a 24-h period. In the mock, seen from both the orthogonal view and transverse cross section view, the expression of the *pMIR171B::H2B-GFP* reporter in the same living SAM was specific in L1 at both 0 h (Fig. 7e) and 24 h after the treatment (Fig. 7f). In contrast, 24 h after the treatment of Dex, the *pMIR171B::H2B-GFP* signal in the living SAM was greatly induced in cells below the L1 layer (Fig. 7g, h). In addition, through RNA in situ hybridization, we also detected the strong induction of *pMIR171B::H2B-GFP* reporter in the SAM and in the young leaves after the Dex treatment of the *pMIR171B::H2B-GFP; 35S::ATML1-GR* plant (Supplementary Fig. 14). The

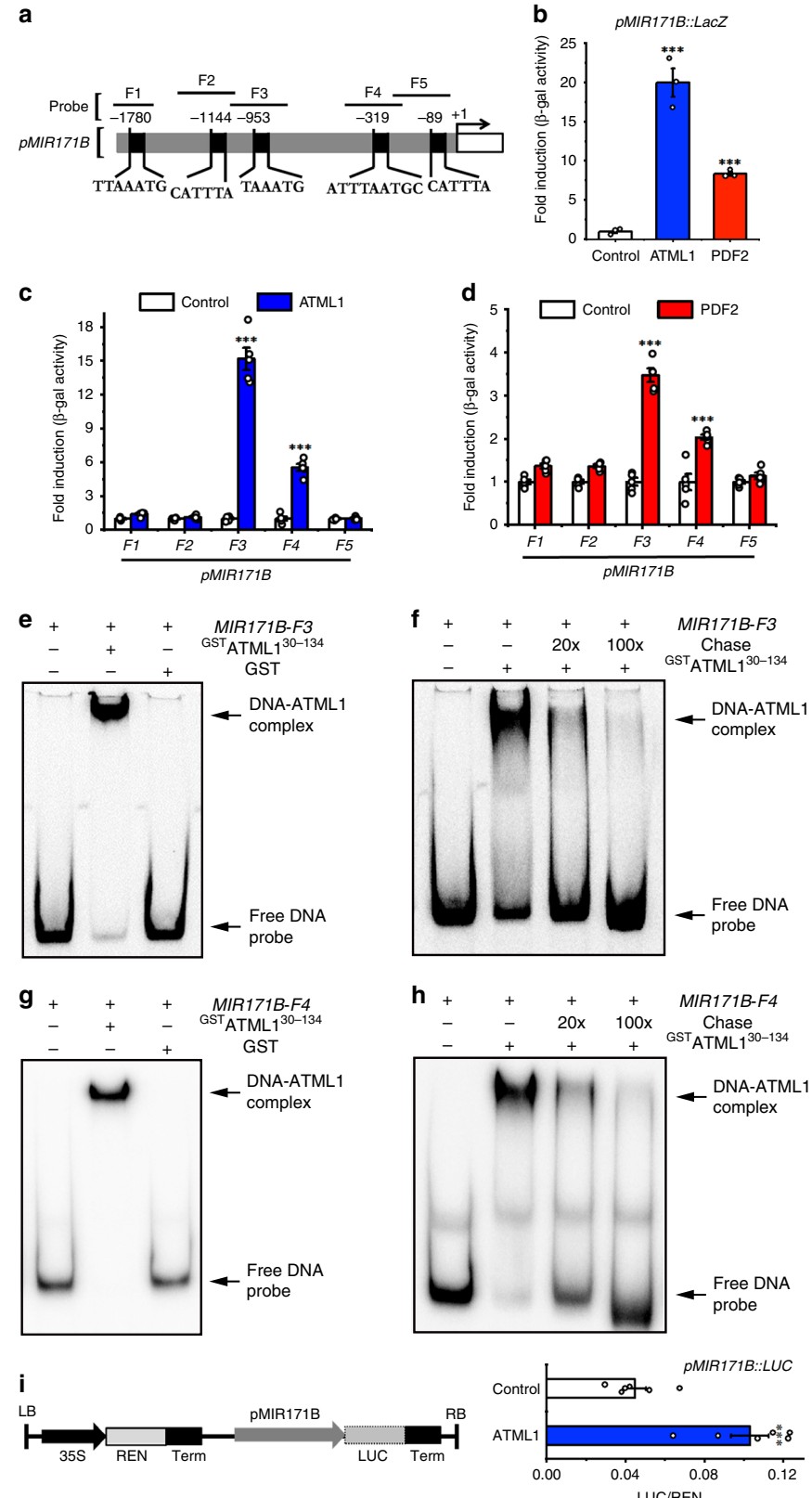

induced expression pattern (Supplementary Figs. 13 and 14) is comparable to the expression of *ATML1* in the same *pMIR171B::H2B-GFP; 35S::ATML1-GR* line (Supplementary Fig. 12). Interestingly, using the similar procedure, we found that the *pMIR171A::H2B-GFP* reporter was also induced upon Dex treatment of the *35S::ATML1-GR* line, with strong activation in young leaves but only mild and uneven induction in the SAM (Supplementary Fig. 15), suggesting potential differences in the regulation of the *MIR171A* promoter and the *pMIR171B* promoter in distinct cell types. In parallel, we found the transcript levels of *GFP*, *MIR171A*, and *MIR171B* in the *35S::ATML1-GR; pMIR171B::H2B-GFP* plants were significantly increased in

**Fig. 3 ATML1 and PDF2 directly activate *MIR171B*. a** Schematic structure of the *MIR171B* promoter. The expanded diagram shows the sequences of putative L1 boxes on the promoter. Probes indicate the DNA fragments used in the EMSA and Y1H. +1 indicates the transcription start site. **b** The full-length promoter of *MIR171B* interacts with ATML1 and PDF2 proteins in Y1H. Y axis: the relative activity of the *pMIR171B::lacZ* reporter. All the numbers are normalized to the average value of the empty vector control. Bars: mean ± SE ($n = 3$ biological replicates). ***$P < 0.001$ (Student's two-tailed $t$-test). **c**, **d** The *MIR171B-F3* and *MIR171B-F4* fragments from the *MIR171B* promoter interact with ATML1 (**c**) and PDF2 (**d**) proteins in Y1H. Y axis: the relative activity of each *lacZ* reporter when the empty vector control, ATML1 (**c**) or PDF2 (**d**) is present. All the numbers are normalized to the average value of the empty vector control. Bars: mean ± SE ($n = 5$ biological replicates). ***$P < 0.001$ (Student's two-tailed $t$-test). **e**–**h** EMSA shows that $^{GST}$ATML1$^{[30-134]}$ but not $^{GST}$ binds to Cy5-labeled *MIR171B-F3* probe (**e**) and *MIR171B-F4* probe (**g**). The $^{GST}$ATML1$^{[30-134]}$-induced mobility shift of the Cy5-labeled *MIR171B-F3* probe (**f**), and the Cy5-labeled *MIR171B-F4* probe (**h**) can be chased away when excess amount of the unlabeled DNA probe (chase) is present. Arrows indicate free DNA probes and DNA-protein complexes. **i** Dual-luciferase assays show that ATML1 activates the transcription of *MIR171B*. Left panel: structure of the reporter construct. Right panel: LUC/REN, ratio of LUC to REN activity. The LUC/REN in the cells co-transformed with the reporter *pMIR171B::LUC* and the ATML1 effector ($n = 6$ biological replicates) is significantly higher than that in the cells co-transformed with the same reporter and empty vector control ($n = 6$ biological replicates). Bar: mean ± SE. ***$P < 0.001$ (Student's two-tailed $t$-test). Source data underlying Fig. 3b–i are provided as a Source Data file.

response to the Dex treatment compared with the mock control (Fig. 7i, j, k). Taken together, consistent with the model predictions (Fig. 7c, d, Supplementary Fig. 16), our experiments demonstrated the immediate activation of *MIR171* production by transient activation of ATML1 in vivo (Fig. 7e–l, Supplementary Figs. 12–15), supporting the first step of the proposed L1-miR171-HAM signaling cascade.

To examine the second step of the L1-miR171-HAM signaling cascade, we performed the RNA in situ hybridization of *HAM1* in the SAMs of the *35S::ATML1-GR* plants with either mock or the Dex treatment at the same time using the identical procedure. The expression of *HAM1* mRNA in the mock-treated SAM was consistent with the previous observation[6,7,9], with a concentration gradient from the epidermis to the deep cell layers. The expression of *HAM1* mRNA in the Dex-treated SAM was greatly reduced and was almost undetectable in the meristem and young leaf primordia (Fig. 7o, p). Consistently, the qPCR experiments showed that the expression levels of *HAM1* and *HAM2* were significantly decreased in the Dex-treated *35S::ATML1-GR* plants compared with the mock control (Fig. 7l, Supplementary Fig. 17). These results are consistent with the model prediction (Fig. 7m, n) (Supplementary Movies 14 and 15). Taken together, both the modeling and experimental results support our proposed regulatory pathway, which starts from ATML1, transfers to miR171, and arrives at HAM, suggesting the epidermis-derived signal cascade shapes the concentration gradient of HAM in the SAM.

**Patterns of *ATML1* and *MIR171* in de novo stem cell niches.** In our previous work, we found that the de novo initiation of new meristem from leaf axils[30] also requires the concentration gradient of HAM[9]. Thus, we wondered whether the L1-miR171-HAM signaling cascade is also active in de novo axillary stem cell niches. We first performed the RNA in situ hybridization of *ATML1* in the AMs from early to late stages during axillary meristem initiation (Fig. 8a–d). *ATML1* expression is not detectable at S1 (Fig. 8a). However, it starts to appear in the epidermis as early as when a group of cells forming a bulge in the leaf axils (Fig. 8b), and the expression continues in the epidermis at the stages thereafter (Fig. 8c, d). In parallel, to probe the promoter activity of *pMIR171A* and *pMIR171B*, we performed the RNA in situ hybridizations of the *GFP* in the developing AMs at different stages from either the *pMIR171A::H2B-GFP* plants or the *pMIR171B::H2B-GFP* plants. The expression patterns of *MIR171* from early to late stages are largely comparable to that of *ATML1* (Fig. 8e–l). These results together with our previous work[9] suggest a sequential regulation of gene expression during new meristem initiation: *ATML1* is firstly activated at the early stage when the meristem identity is specified. Then, the miR171

expression is turned on in the epidermis. Consequently, the apical-basal concentration gradient of *HAM* is defined[9], which shapes the *CLV3* expression pattern and promotes the initiation of new stem cell niches[9].

**Discussion**
With a high cellular resolution, we have revealed the expression patterns of all four *MIR171/170* genes in Arabidopsis SAMs, which shows that miR171 originates from the epidermis during shoot development (Fig. 1). *MIR171A* and *MIR171B* genes are highly expressed in epidermis of the SAM (Fig. 1a, b), whereas *MIR171C* and *MIR170* genes are also specifically expressed in epidermis of the SAM but at a much lower level (Fig. 1c, d). These results suggest that these four genes are responsible for the miR171 activity in the SAM and SAM derived leaves and flower meristems, while *MIR171A* and *MIR171B* genes contribute more to the total miR171 level in the SAM. Further, we found that *MIR171* genes are the direct targets of both ATML1 and PDF2 transcription factors (Figs. 2, 3, and 7).

It has been well documented that ATML1 and PDF2 recognize L1 box, the consensus cis-regulatory element, to regulate down-stream targets[22,29]. We found that on the promoters of *MIR171* genes, the (A)TT(A/T)AATG(C/T) sequences mediate the ATML1 binding. DNA sequences containing at least six base pairs of the conserved TAAATG may also function as the ATML1 binding sites, depending on the flanking sequences of each core element. These results are consistent with the previous studies on the sequence specificity and variation of the L1 box[22,24,29]. When two of the TTAAATGC L1 boxes were mutated, the expression of *MIR171A* was undetectable or largely reduced in the epidermis of the SAM, suggesting these L1 boxes are essential for the direct activation of *MIR171* by ATML1/PDF2. Interestingly, the muta-ted *MIR171A* reporter was weakly expressed in the epidermis of flower sepal primordia (Fig. 5), suggesting there exists other cis-regulatory elements and/or transcriptional regulators that can activate *MIR171* expression in developing organs. Indeed, it has been reported that *MIR171* expression is reduced in the *ham* mutants[13,31], and the *MIR171* expression is also regulated by light[32]. Therefore, it is noteworthy to identify other factors con-tributing to the regulation of *MIR171* expression in the future.

We found that the miR171 insensitive transcriptional reporter (*pHAM2::H2B-GFP*) was expressed in all layers of the SAM. However, the miR171 sensitive translational reporter (*pHAM2::YFP-HAM2*) was not detectable in the L1 layer, largely reduced in the L2 layer, and moderately reduced in upper corpus (Fig. 1e, f). These results suggest that miR171 moves across at least two cell layers within the SAM, leading to a complete repression of HAM in L1, strong repression in L2, and moderate repression in cells located at upper corpus. Therefore, miR171 in SAMs serves as a

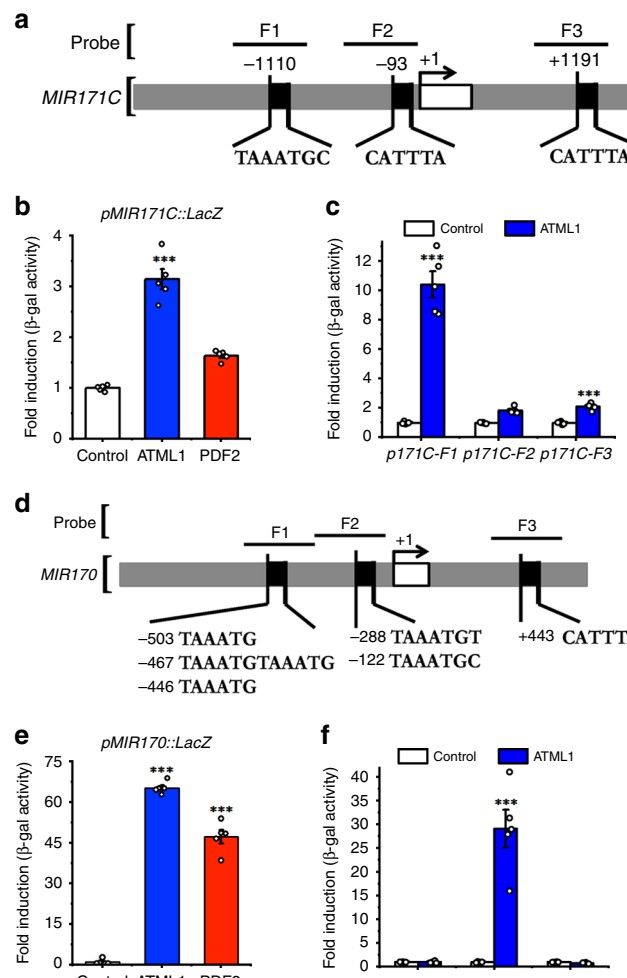

**Fig. 4 ATML1 binds to the promoters of *MIR171C* and *MIR170*.**
**a** Schematic structure of the *MIR171C* promoter. The expanded diagram shows the sequences of putative L1 boxes on the promoter. Probes indicate the DNA fragments used in the Y1H assay. +1 indicates the transcription start site. **b** The full-length promoter of *MIR171C* interacts with ATML1 protein in Y1H. Y-axis: the relative activity of the *pMIR171C::lacZ*. All the numbers are normalized to the average value of the empty vector control. Bars: mean ± SE ($n = 5$ biological replicates). ***$P < 0.001$ (Student's two-tailed *t*-test). **c** The *MIR171C-F1* fragment from the *MIR171C* promote interacts with the ATML1 protein in Y1H. Y-axis: the relative activity of the *pMIR171C-F1::lacZ*, *pMIR171C-F2::lacZ* and *pMIR171C-F3::lacZ* reporters. All the numbers are normalized to the average value of the empty vector control. Bars: mean ± SE ($n = 5$ biological replicates). ***$P < 0.001$ (Student's two-tailed *t*-test). **d** Schematic structure of the *MIR170* promoter. The expanded diagram shows the sequences of putative L1 boxes on the promoter. Probes indicate the DNA fragments used in the Y1H assay. +1 indicates the transcription start site. **e** The full-length promoter of *MIR170* interacts with ATML1 protein in Y1H. Y axis: the relative activity of the *pMIR170::lacZ* reporter. All the numbers are normalized to the average value of the empty vector control. Bars: mean ± SE ($n = 5$ biological replicates). ***$P < 0.001$ (Student's two-tailed *t*-test). **f** The *MIR170-F2* fragment from the *MIR170* promoter interacts with the ATML1 protein in Y1H. Y-axis: the relative activity of each *lacZ* reporter. All the numbers are normalized to the average value of the empty vector control. Bars: mean ± SE ($n = 5$ biological replicates). ***$P < 0.001$ (Student's two-tailed *t*-test). Source data underlying Fig. 4b, c and Fig. 4e, f are provided as a Source Data file.

mobile signal, and it shapes the HAM patterns not only in epidermis but also in inner cell layers. The movement of miRNA through limited distance in SAMs is not surprising, and it has been reported previously[33,34]. For example, when expressed in tunica (L1 and L2) layers, the movement of an artificial miRNA is limited and it only moves down one cell layer[33]. As another example, miR394b that is transcribed in the L1 layer of the SAM can form a concentration gradient with the movement through one or two cell layers[34]. Our finding together with these reports support the idea that the movement of microRNA through limited number of cell layers in the SAMs is likely an important feature for microRNAs to function. It will be interesting to further investigate the regulatory mechanism underlying the movement of miR171 among different cell layers.

With the aid of a computational model, we established and simulated the key linkage from the epidermal signal (ATML1) to the concentration gradient of *HAM* in 3D SAMs. The transient activation of ATML1 results in quick and strong induction of *MIR171* and the subsequently reduced *HAM* expression in the shoot, which was illustrated by both model simulation and live cell imaging. In the de novo axillary meristem, results presented here together with our previous work suggest that L1-miR171-HAM regulatory cascade is also active during the development of new axillary meristems (Fig. 8). ATML1 is not universally expressed at the epidermis throughout the whole plant body[20,21], nor it is turned on at the leaf axis until a group of cells have acquired meristematic identity and start to form bulge (Fig. 8a, b). When ATML1 is expressed, *MIR171* starts to express at the epidermis of initiating meristems and remains its expression afterwards. This finding provides additional evidence supporting that ATML1 activates miR171, and it suggests that such regulation plays a role in the establishment of the concentration gradient of HAM and likely the apical-basal polarity of *CLV3* expression as well[9].

The L1-miR171-HAM signaling cascade we proposed and validated in this study is specific to the SAMs and SAM derived young above-ground tissues, and all four *MIR171/170* genes are not expressed at the epidermal layer of the roots, from the root cap to the differentiation zones (Supplementary Fig. 18). In contrast, the reporters of *MIR171B* and *MIR171C* are expressed in the inner layer at the differentiation zone in the root (Supplementary Fig. 18) (Supplementary Movies 16 and 17). Thus, the function and regulation of the miR171 family during the root development is still an open question.

In summary, our work here uncovers the L1-miR171-HAM signal cascade during the shoot development. In the SAMs, this signal cascade determines the concentration gradient of HAM proteins, which are not present in epidermis but are regulated by signals initiated from the epidermis. Future study linking the L1-miR171-HAM signal cascade to various developmental processes mediated by HAM, including the previously proposed WUS-HAM-CLV3 regulatory loop, will provide a comprehensive view of this regulatory circuit in defining the plant shoot architecture.

## Methods

**Plant materials and growth conditions.** *Arabidopsis thaliana* plants were grown in SunGro horticulture propagation mix soil under short-day conditions (8 h light/16 h dark cycle) at 22 °C, or continuous light at 22 °C. The *atml1-1/+ pdf2-1* double mutant was previously described[7,23]. For the genotyping of *atml1-1 pdf2-1*, a primer specific for the T-DNA left border (LB 5′- CATTTTTATAATAACGCTGCGG ACATCTAC-3′) was used in tandem with *ATML1*-specific primers (5′-CCGG CTTGTGATCTTTCCG-3′ and 5′- GGCTCCGTCGCAGGCCAGAGC-3′) or *PDF2*-specific primers (5′-ATTGATAGGATCTCTGCTATTG-3′ and 5′-CACAT CCATGAGAATCTCAACGA-3′).

**Plasmid constructions for transgenic plants.** *H2B-GFP* was described previously[35]. To generate *pMIR171A::H2B-GFP*, 457 bp *MIR171A* 3′ terminator were

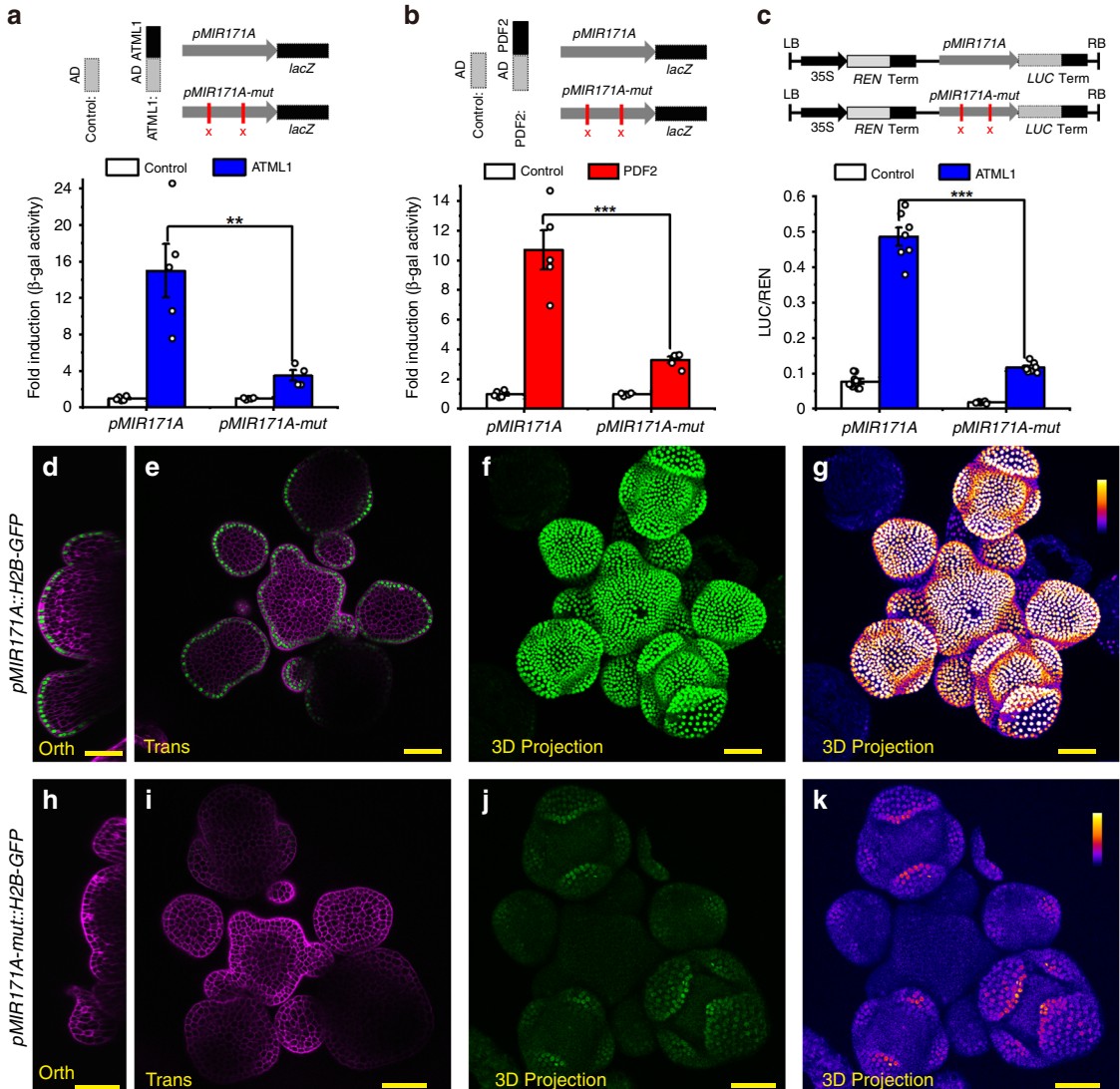

**Fig. 5 Epidermis specific cis-regulatory elements are required for the activation of miR171. a, b** Mutations in the *MIR171A* promoter significantly reduced its interaction with ATML1 (**a**) or PDF2 (**b**) in Y1H. Upper panel: sketch of the reporter used in the study. AD with no fusion protein was included as the control. The *lacZ* reporter was under the control of either the *MIR171A* promoter (*pMIR171A::lacZ*) or the *MIR171A* promoter with two sites of TTAAATGC mutated (*pMIR171A-mut::lacZ*). Lower panel: mutations in *pMIR171A* significantly reduced the activation of the *pMIR171A::lacZ* by ATML1 (**a**) or PDF2 (**b**). Bar: mean ± SE (**a**: $n = 5$ biological replicates for AD control and AD-ATML1, except that $n = 4$ biological replicates for AD-ATML1 on *pMIR171A-mut::lacZ*; **b**: $n = 5$ biological replicates for AD control and AD-PDF2). **$P < 0.01$, ***$P < 0.001$ (Student's two-tailed $t$-test). **c** Dual-luciferase assay shows that the activation of *MIR171A* promoter activities by ATML1 is greatly compromised by the two mutations in the *MIR171A* promoter. The LUC/REN in cells co-transformed with the *pMIR171A::LUC* reporter and the ATML1 effector ($n = 7$ biological replicates) is significantly higher than that in tobacco cells co-transformed with the *pMIR171A-mut::LUC* reporter and the same effector ($n = 8$ biological replicates). ***$P < 0.001$ (Student's two-tailed $t$-test). **d–g** Expression of *pMIR171A::H2B-GFP* in an *Arabidopsis* SAM, from the orthogonal view (**d**), in the transverse optical section in corpus (**e**), and from the 3D projection view (**f, g**). **h–k** Expression of *pMIR171A-mut::H2B-GFP* in the SAM, from the orthogonal view (**h**), transverse optical section view in corpus (**i**), and from the 3D projection view (**j, k**). The confocal settings for imaging GFP in **d–k** are identical, and the quantification of GFP is indicated by the identical color bar in **g, k. d, e, h, i**: merge of GFP (green) and PI counterstain (purple); **f, j**: GFP (green); **g, k**: GFP quantified from **f, j** (quantification indicated by color). Scale bar: 20 μm; color bar in **g, k**: fire quantification of the signal intensity. Source data underlying Fig. 5a–c are provided as a Source Data file.

amplified from Col-0 genomic DNA with primers 5′-TACAggcgcgccTCAGTAC TCTCTCGTCTCTATTTT-3′ and 5′-TACAggcgcgccTGATTGTGACATTTTAC ATTGCTTC-3′ (restriction enzyme sites are in lower case) and cloned at 3′ of the *H2B-GFP* fragment to generate *H2B-GFP-MIR171A-3′ terminator*. Then 1474 bp *MIR171A* promoter was amplified using the primers 5′-ACAAgcggccgcGATACT CCACTTTTAGGCTCCATCTT-3′ and 5′-ACAAgcggccgcAAAGGGACTCTCTC ATGCTTAAAG-3′ and inserted at the NotI site in front of the *H2B-GFP-MIR171A-3′ terminator* to get *pMIR171A::H2B-GFP-MIR171A-3′ terminator*. *pMIR171A::H2B-GFP-MIR171A-3′ terminator* fragment was then cloned into the binary vector pMOA34.

For the construction of *pMIR171B::H2B-GFP*, 1852 bp *MIR171B* promoter was amplified with 5′-ACAAgcggccgcATATAAAACATGCTATTGCTTCTT-3′ and 5′-ACAAgcggccgcTAAAACCACTCTTGTTCGACTATAATC-3′ and cloned in front of H2B-GFP; and 960 bp 3′ terminator of *MIR171B* was PCR amplified with 5′-TACAggcgcgccAAGATAGTTATTATAACCTTAAAG-3′ and 5′-TACAg gcgcgccGAGTCCTTATTGTTGTGCCTTTTTA-3′ and cloned 3′ of the H2B-GFP, then the fused DNA fragment was cloned into pMOA34.

For the construction of *pMIR171C::H2B-GFP*, 2157-bp *MIR171C* promoter was amplified with 5′-ACAAgcggccgcTAATTCAAACCGAATTAGACCAAAA-3′ and 5′-ACAAgcggccgcTCGACTCTTCAGTTGCTTATTACACCA-3′ and cloned in

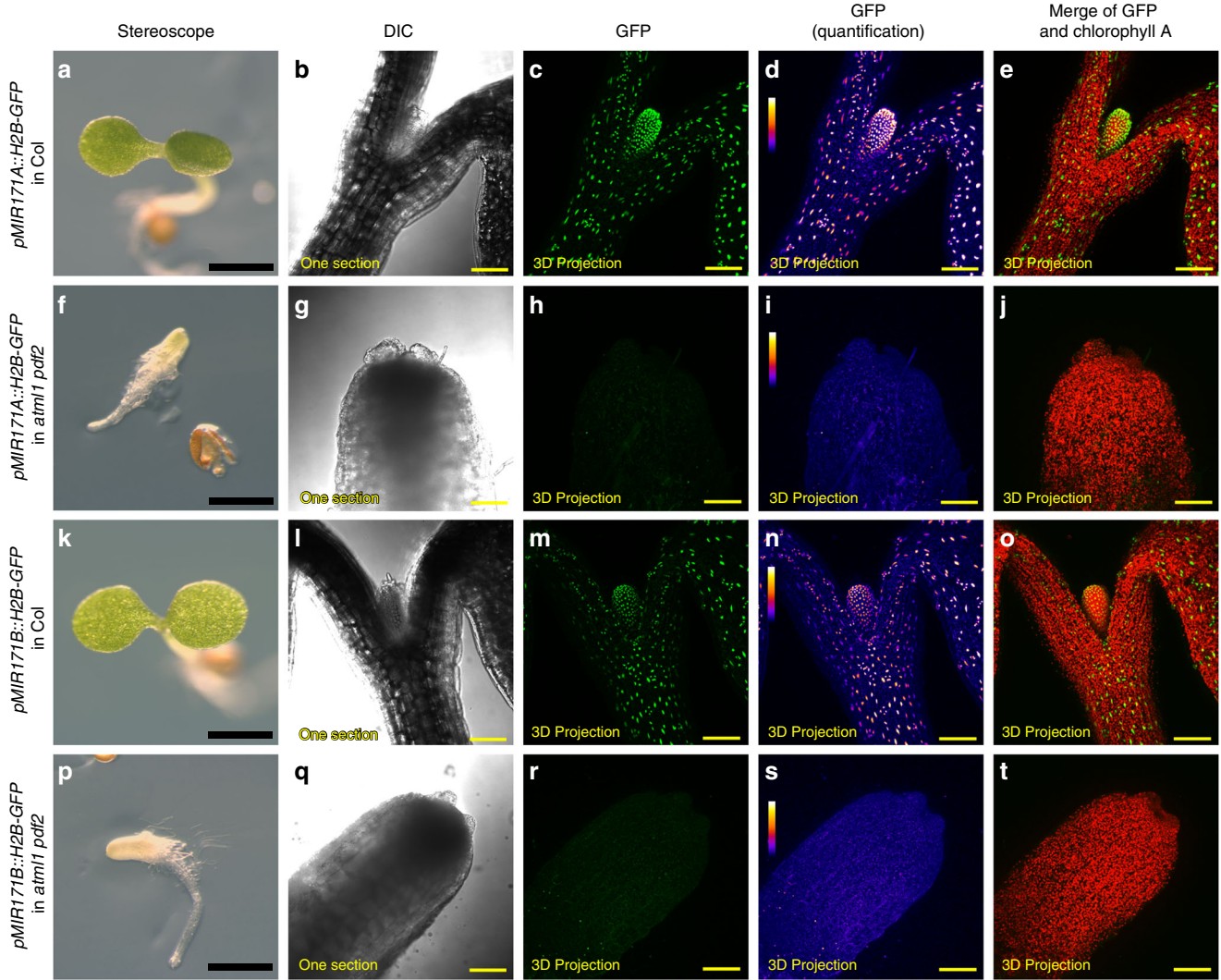

**Fig. 6 Epidermal specification pathway is required for the expression of *MIR171* genes. a–e** The wild type seedling (Col) expressing the *pMIR171A::H2B-GFP* reporter. **f–j** The *atml1 pdf2* double mutant seedling expressing the *pMIR171A::H2B-GFP* reporter. **k–o** The wild type seedling (Col) expressing the *pMIR171B::H2B-GFP* reporter. **p–t** The *atml1 pdf2* double mutant expressing the *pMIR171B::H2B-GFP* reporter. Panels (from left to right): seedlings used for the confocal imaging; DIC from one optical section; GFP (green) in 3D projection view, quantified GFP (quantification indicated by color) in 3D projection view; merge of GFP (green) and chlorophyll (red) in 3D projection view. Scale bars: 1 mm (**a**, **f**, **k**, **p**) and 100 μm (**b–e**, **g–j**, **l–o**, and **q–t**); color bar (**d**, **i**, **n**, **s**): fire quantification of the signal intensity.

front of H2B-GFP; and 1161 bp 3′ terminator of *MIR171C* was PCR amplified with 5′-TACAggcgcgccAATAGTTTAAAGATTCTATGTTAGTTG-3′ and 5′-TACA ggcgcgccTGTAATTAAGATCTGCGACCACTTC-3′ and cloned at 3′ of the H2B-GFP fragment, then the fused DNA fragment was introduced into pMOA34.

For the construction of *pMIR170::H2B-GFP*, 1251 bp *MIR170* promoter was amplified with 5′-ACAAgcggccgcTCCTAACCTGTTCCTTTTGA-3′ and 5′-AC AAgcggccgcGAAACATAGTGTATCTGTTTTGTAA-3′ cloned in the front of H2B-GFP; and 582 bp 3′ terminator of *MIR170* was PCR amplified with 5′-TACA ggcgcgccTCGCTTCTCTCGTATTTTGAA-3′ and 5′-TACAggcgcgccGCTCCACC ACAAAGCTCTTC-3′ and cloned at 3′ of the H2B-GFP fragment, then the fused DNA fragment was introduced into pMOA34.

The promoter and 3′ terminator of *pHAM2::H2B-GFP* were the same as that in the *pHAM2::YPET-HAM2* construct described before[9]. The above described constructs were transformed into L*er* through floral dip[36], except the *pHAM2:: YPET-HAM2* construct, which was transformed into *ham123* triple mutants. The *pMIR171A::H2B-GFP* and *pMIR171B::H2B-GFP* constructs were also transformed into Col. The expression patterns of each construct were confirmed in multiple lines.

To construct of *pMIR171A-mut::H2B-GFP*, two L1 boxes in the *MIR171A* promoter were mutated from TTAAATGC to CCCGGTGC (Supplementary Fig. 5) using two round of overlapping PCR[37]. For the first round of overlapping PCR, the primers 5′- AGTAGTAGCTTAATATCAATGTCGCACCGGGGGAACATT ATAAGCTACTAATATGAAA-3′, and 5′-TTTCATATTAGTAGCTTATA ATGTTCCCCCGGTGCGACATTGATATTAAGCTACTACT-3′ were used to

mutate the L1 box located at distal end (F1) of *MIR171A* promoter. For the second round of overlapping PCR, the primers 5′-AGGAAATGGAAGGTATGGAGCA CCGGGGACCTCATCTACTACCAAAGCACAAA-3′ and 5′-TTTGTGCTTTGG TAGTAGATGAGGTCCCGGTGCTCCATACCTTCCATTTCCT-3′ were used to mutate the L1 box located at proximal end (F3) of *MIR171A* promoter. The mutations have been confirmed through the sequencing. *pMOA34 pMIR171A-mut: H2B-GFP* was transformed to L*er* through flower dipping[36].

To generate *35S::ATML1-GR*, ATML1 cDNA was amplified using the primers 5′- CATAgagctcATGTATCATCCAAACATGTTCGA-3′ and 5′-AGTTgagctcTG GTGGTGCTGCTGCTGCTGCTGCGGCTCCGTCGCAGGCCAGAGCGG-3′ (restriction enzyme sites are indicated in lower case and underlined is the linker) into the SacI site of pGreen0029 35S::GR. *pGreen0029 35S::ATML1-GR* was transformed into the *pMIR171A::H2B-GFP* line 14 and the *pMIR171B::H2B-GFP* line 3 in L*er* background.

**Confocal live imaging**. All of the florescence reporters were imaged using a Zeiss LSM 880 upright confocal microscope. The live-imaging experiments were performed as previously described[7,38,39]. The shoot apices including SAMs were imaged with the water dipping lens (Zeiss). To image GFP and PI simultaneously in the SAMs and roots or to image GFP and chlorophyll simultaneously in the young seedlings, the GFP, chlorophyll or PI was excited using a 488-nm laser line. To image YPET and PI simultaneously in the SAMs or YPET and chlorophyll

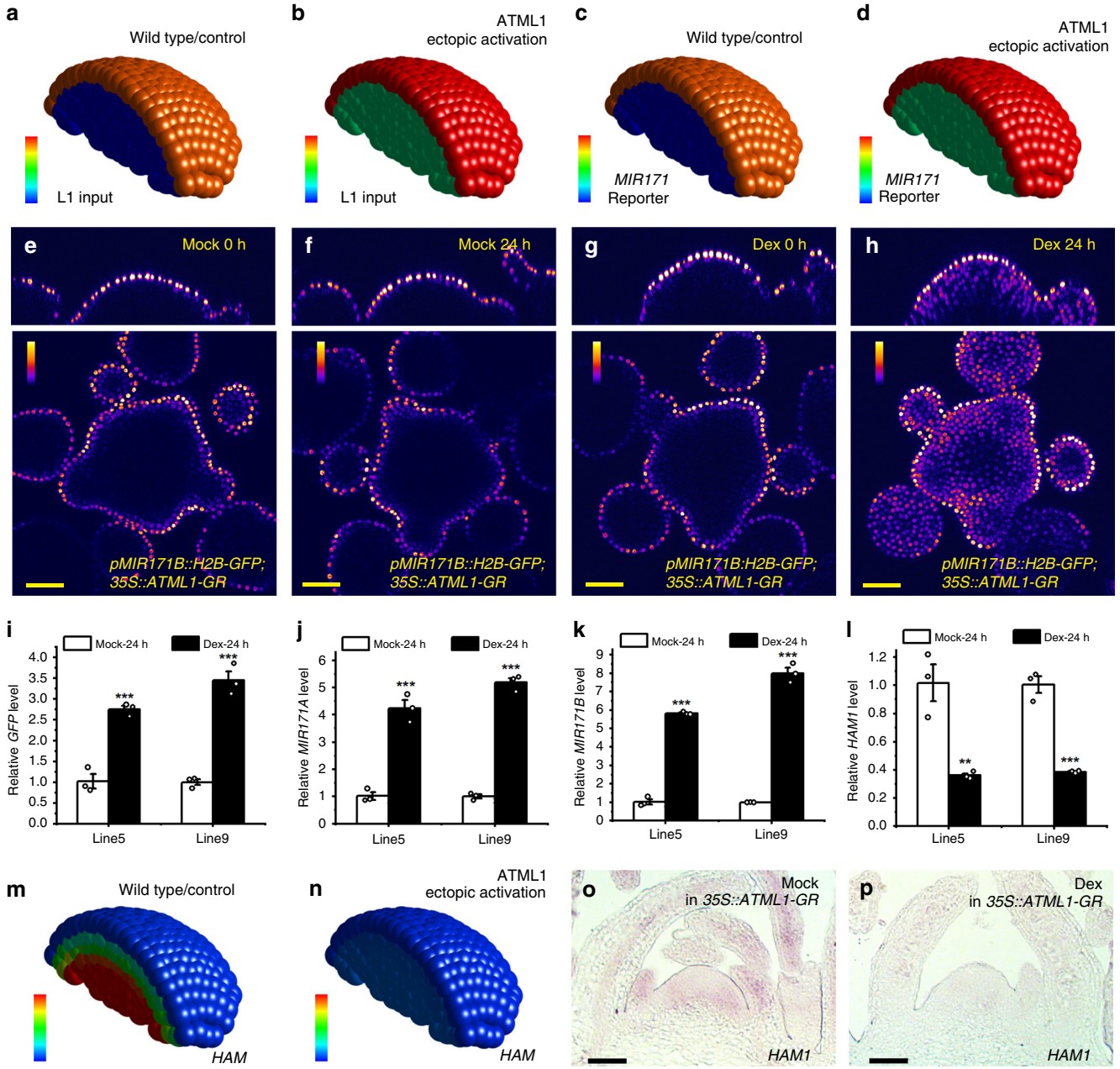

**Fig. 7 The 3D computational model simulates the L1-miR171-HAM regulatory circuit in SAMs. a, b** Input of the ATML1 level (L1 input) for the computational simulation in (**c, m**) and (**d, n**). Color indicates the level of ATML1 input, with the gradient from red (maximum, 1.1 a.u.) to blue (none). **c, d** Simulated promoter activity of *MIR171* in the SAMs from the wild type control (**c**) and the plant with ectopic activation of ATML1 (**d**). Color indicates the simulated *MIR171* reporter activity, with the gradient from red (maximum, 1.1 a.u.) to blue (none). **e–h** Confocal live imaging of the *pMIR171B::H2B-GFP* reporter in the SAMs of *35S::ATML1-GR* plants, with the mock control (**e, f**) and the Dex treatment (**g, h**). Using identical imaging settings, the same SAM was imaged at 0 h and 24 h after the indicated treatment. Colors indicate quantified GFP intensities, and color bars represent fire quantification of signal intensity. Panels: (top) orthogonal view; (bottom) transverse optical section view from corpus. **i–l** Quantification of the gene expression for *GFP* (**i**), *MIR171A* (**j**), *MIR171B* (**k**), and *HAM1* (**l**) in two independent *35S::ATML1-GR; pMIR171B::H2B-GFP* transgenic lines (#5 and #9) 24 h after the mock or Dex treatment. Bars: mean ± SE. **P < 0.01, ***P < 0.001 (Student's two-tailed *t*-test). *n* = 3 biological replicates. **m, n** Simulated *HAM1* mRNA expression patterns in SAMs from the wild type control (**m**) and the plant with ectopic activation of ATML1(**n**). Color indicates the simulated *HAM1* mRNA levels, with the gradient from red (at or above 2 a.u.) to blue (none). **o, p** RNA in situ hybridizations for *HAM1* mRNA in the SAMs of *35S::ATML1-GR; pMIR171B::H2B-GFP* plants 5 days after the mock (**o**) or Dex (**p**) treatment. One set of representative results are shown, and similar results from three independent biological replicates were obtained for each treatment. Scale bar: 50 μm (**o–p**). The template of a whole SAM was used for all the simulations, and only a half of the SAM is displayed for visualization of cells in inner layers. Source data underlying Fig. 7i–l are provided as a Source Data file.

simultaneously in the young seedlings, YPET, chlorophyll or PI was excited by a 514-nm laser line.

To image *pMIR171A::H2B-GFP*, *pMIR171B::H2B-GFP*, *pMIR171C::H2B-GFP*, *pMIR170::H2B-GFP*, *pHAM2::H2B-GFP* and *pHAM2::YPET-HAM2* reporters, the

shoots from 9-day-old seedlings grown in short days and the inflorescence shoot apices from the bolted plants were used (Fig. 1).

To examine the expression of *MIR171A* and *MIR171B* in the *atml1-1 pdf2-1* double mutant, the *pMIR171A::H2B-GFP* reporter (line 11) and the *pMIR171B::*

Early ➤ Late

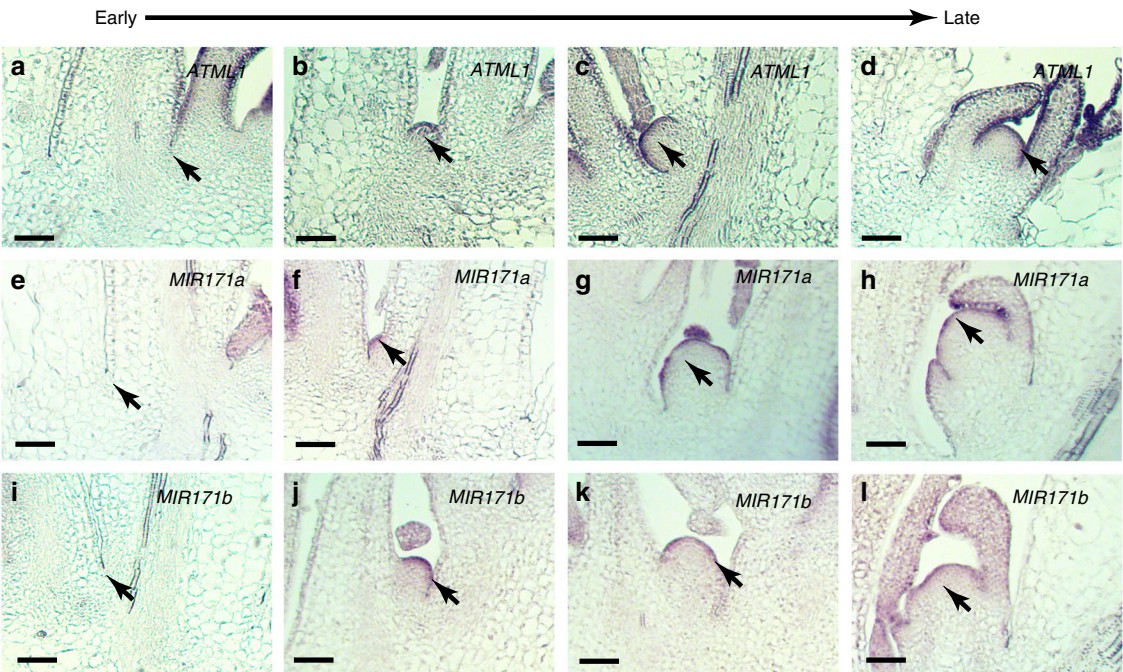

**Fig. 8 Pattern formation of L1 and miR171 in de novo axillary stem cell niches. a–d** RNA in situ hybridization of *ATML1* mRNA in wild-type (L*er*) plants, from early to late developmental stages during the de novo formation of the axillary meristem (AM). **e–h** RNA in situ hybridization of *GFP* mRNA in *pMIR171A::H2B-GFP* plants showing the promoter activity of *MIR171A*, from early to late stages of AM initiation. **i–l** RNA in situ hybridization of *GFP* mRNA in *pMIR171B::H2B-GFP* plants showing the promoter activity of *MIR171B*, from early to late stages of AM initiation. The results have been confirmed by three biological replicates. Scale bar: 50 μm.

*H2B-GFP* reporter (line 5) in Col background were crossed into *atml1-1/+ pdf2-1* mutants[23]. Within the F2 population, the *atml1-1/+ pdf2-1* lines were identified and maintained through the PCR based genotyping and the expression of the reporter was confirmed through confocal imaging. Multiple F3 generation plants of *pMIR171A::H2B-GFP* in *atml1-1/+ pdf2-1* and *pMIR171B::H2B-GFP* in *atml1-1/+ pdf2-1* background were first grown on hygromycin plates to identify the lines containing the homozygous reporter. Then the identified lines with the homozygous reporter were grown on half MS media for 5 days in short day. The seedlings with the previously described *atml1-1 pdf2-1* phenotype (no cotyledons)[23] were identified as the double homozygous mutants and imaged together with the homozygous reporter in wild type using confocal microscope, and several individual seedlings of these *atml1-1 pdf2-1* double mutants have been confirmed through the genotyping PCR.

To investigate the expression of *MIR171B* in response to the transient activation of ATML1 in Arabidopsis, the *35S::ATML1-GR; pMIR171B::H2B-GFP* plants were grown in short days for 25 days and then moved to continuous light to induce flowering. The inflorescence shoot apices were dissected out and mounted in the GM media supplemented with B5 vitamins, and the SAMs were live-imaged at both 0 hour and 24 h after mock or the 10-μM Dex treatment. The live imaging results in this study have been confirmed using two independent *35S::ATML1-GR; pMIR171B::H2B-GFP* lines (lines 5 and 9) with more than three biological replicates from each line. All the images were processed and quantified in Fiji.

**Protein expression and purification.** GST or $^{GST}$ATML$^{30-134}$ proteins were expressed and purified following the same procedure as previously reported[7,40]. Briefly, an empty pGEX vector or a pGEX vector carrying the open reading frame of ATML$^{30-134}$ was transformed in Rosetta E. coli cells. Expression of the recombinant proteins were induced with 0.2 mM IPTG at 16 °C overnight. The expressed GST or $^{GST}$ATML$^{30-134}$ proteins were extracted from the E. coli cells using Glutathione Sepharose resins (GE healthcare). Proteins eluted from the resins were further purified using size exclusion chromatography, and eluate fractions containing the protein of interest were combined and used for the study (Supplementary Fig. 2).

**Gel mobility shift assay (EMSA).** Cy5-labeled DNA fragments were used as probes for EMSA. The labeled DNA probes were synthesized through regular PCR using oligos conjugated with Cy5 at the 5′ end (synthesized and purified by IDT). The DNAs were then purified (Qiagen) and stored in amber tubes at −20 °C before use. The *MIR171A* probe F1 (Fragment 1) (Fig. 2a, f, g) was amplified from Col-0 genomic DNA using the primers 5′-ATGTGAATGAAGCCAAGAATCA-3′ and 5′-TTTCTTTTCAGGATTACTGTTTTCG-3′. The *MIR171A* probe F2 (Fragment 2) (Fig. 2a, h, i) was amplified from Col-0 genomic DNA using the primers 5′-GG

GTTCATAACTTGCTTGAGG−3′ and 5′-TTTCCTTCTTGAGGAGGTTCA-3′. The *MIR171A* probe F3 (Fragment 3) (Fig. 2a, j, k) was amplified from Col-0 genomic DNA using the primers 5′-CCTCCTCAAGAAGGAAAGACAG-3′ and 5′-AGTGAGAATGTCTGTGGGGAGT-3′. The *MIR171B* probe F3 (Fragment 3) (Fig. 3a, e, f) was amplified from Col-0 genomic DNA using the primers 5′-ATGTT GATTACCCATATACTTTGCT-3′ and 5′-TCCCACGAGCTTTCAATTTAGT-3′. The *MIR171B* probe F4 (Fragment 4) (Fig. 3a, g, h) was amplified from Col-0 genomic DNA using the primers 5′-CCTGATAATTTGATATCATCGGTTG-3′ and 5′-ATGCGGCCAGTAGCAAAGT-3′.

When assaying the gel mobility shift, 7.5 nM Cy5-labeled DNA probe in buffer containing 15 mM Tris (pH 7.5), 50 mM NaCl, 0.1 mM DTT, 5% glycerol, 0.1 mM MgCl$_2$, and 50 ng/μl carrier DNA (Invitrogen) was first incubated with GST or $^{GST}$ATML$^{30-134}$ protein at room temperature for 30 min. The mixture was subsequently fractionated on a 6% DNA Retardation Gel (ThermoFisher Scientific) in running buffer containing 0.5x TBE at 4 °C for 1 h with 100 V power supplied. Gels were scanned by a Typhoon FLA 9500 gel imaging scanner (GE Healthcare) to detect the Cy5-labeled DNA probe. To confirm the interaction between the DNA probe and the $^{GST}$ATML$^{30-134}$ protein, a chase assay was performed by adding excess amount of unlabeled DNA with identical sequence to the fluorescent probe. The unlabeled DNA was added together with the labeled DNA probe before the addition of the $^{GST}$ATML$^{30-134}$ protein. The subsequent steps were the same as described above. All the EMSA experiments in this study were repeated two times and they showed the same results.

**RNA in situ hybridization.** RNA in situ hybridization were performed as previously described[9]. Plant samples were fixed in 4% paraformaldehyde in phosphate-buffered saline (PBS) at 4 °C overnight. Then the samples were embedded in Paraplast X-Tra (McCormic Scientific) and sectioned with 8 μm thickness using a microtome (Thermo Scientific). Sections were transferred to microslides and dried overnight at 42 °C on a slide warmer. Sections were dewaxed and treated at 0.2 M HCl for 20 min, followed by 2 x SSC for 15 min at 70 °C. Then the sections were treated with Protease K (1 μg/ml) for 30 min, followed by 0.1 M thiethanolamine (pH 8.0) and 0.5% acetic anhydride for 10 min. Probes were mixed with the hybridization solution (50% formamide, 300 mM NaCl, 10 mM Tris-pH7.5, 1 mM EDTA, 1x Denhardts solution, 10% Dextran Sulfate, 70 mM DTT, 150 μg/ml tRNA). Samples were hybridized with the indicated probe in the hybridization solution at 53 °C overnight, and the sample slides were washed with 0.2 x SSC four times (30 min each wash) at 53 °C. Thereafter, slides were blocked with 1% Blocking Reagent (Roche) in PBS containing 0.3% Triton X-100 for 45 min, followed by incubation with 1% bovine serum albumin (BSA) in PBS containing 0.3% Triton X-100 for 45 min. Immuno-detection of the hybridized probe was performed by incubating slides with the 0.5% BSA/PBS/0.3% Triton

X100 solution containing the anti-DIG antibody (1:1000 dilution) for 3 h. After that, the slides were washed with PBS and buffer containing 100 mM Tris pH 9.5, 100 mM NaCl, and 50 mM MgCl$_2$. For signal visualization, slides were developed using Western blue stabilized substrate for alkaline phosphatase (Promega).

The probe for *GFP* was cloned into pGEM-T easy (Promega) using the primers: 5′-CAAGGGCGAGGAGCTGTT-3′ and 5′-CTTGTACAGCTCGTCCATGC-3′. The probe for *HAM1* was described previously[9]. The probe for *ATML1* was cloned into pGEM-T easy (Promega) using the primers: 5′-ACTGGCGTAGCAGG GAACTA-3′ and 5′-GGCTCCGTCGCAGGCCAGAGC-3′. Probes were in vitro transcribed in the presence of DIG-UTP using the Sp6/T7 transcription kit (Roche). The synthesized probes were hydrolyzed in Carbonate hydrolysis buffer (60 mM Na$_2$CO$_3$ and 40 mM NaHCO$_3$) at 60 °C.

To investigate the expression patterns of *GFP* in the SAMs of the *35S::ATML1-GR; pMIR171::H2B-GFP* plant with mock or the Dex treatment, the *35S::ATML1-GR; pMIR171::H2B-GFP* plants were grown in short days for 24 days. Then they were sprayed with mock (no Dex) or with 10 μM Dex solution. At 24 h after mock or the Dex treatment, vegetative SAMs were collected for RNA in situ hybridization to *GFP*. This experiment was confirmed with three biological replicates.

To investigate the expression patterns of *GFP* and *HAM1* in the SAMs of the *35S::ATML1-GR; pMIR171::H2B-GFP* plant with mock or the Dex treatment, the *35S::ATML1-GR; pMIR171::H2B-GFP* plants (line 9) were grown in short days for 21 days. Then they were sprayed with mock (no Dex) or with 10 μM Dex solution every the other day. At 5 days after mock or the Dex treatment, vegetative SAMs were collected for RNA in situ hybridization to *GFP* and *HAM1*. RNA in situ hybridization to *GFP* was confirmed with two biological replicates, and RNA in situ hybridization to *HAM1* was confirmed with three biological replicates.

To investigate the localization of miR171 in the wild type SAMs, the DIG labeled miRCURY Locked Nucleic Acid (LNA) miRNA detection probe (5′-CG TGATATTGGCACGGCTCA-3′) of miR171b (Qiagen) was used. The DIG labeled miRCURY LNA miRNA detection probe (5′-TCAAGGTCCGCTGTGAACAC-3′) of miR124 (Qiagen), which specifically recognizes murine miR124, was used as the negative control as previously suggested[41,42]. The RNA in situ hybridization was performed using our standard protocol of RNA in situ hybridization for mRNAs, except that the LNA probes for miR171 and for miR124 were prepared in 50% formamide and heated for 3 min at 85 °C following the same procedure (step 11) described in the published protocol[41]. Both LNA probes were used with the concentration of 100 nM and at 53 °C hybridization temperature. Inflorescences were collected from Ler wild type plants that were grown in short days for 28 days and then continuous light for 17 days (Supplementary Fig. 8). These experiments were repeated with four biological replicates for each probe at the same time with the identical procedure. To harvest vegetative SAMs, Ler wild type plants were grown in short days (Supplementary Fig. 9). These experiments were repeated with five independent biological replicates.

To investigate the expression patterns of *ATML1* in the SAMs of the Ler wild type and the *35S::ATML1-GR; pMIR171B::H2B-GFP* plant (line 9), the plants were grown in short days for 23 days. RNA in situ hybridization to *ATML1* was performed using the identical procedure. This experiment was confirmed with three biological replicates for each genotype.

To investigate the expression pattern of *ATML1* during AM (axillary meristem) initiation, wild type Ler plants were grown in short days for 28 days and then in continuous light for 1–2 days. All these experiments were repeated with three biological replicates.

To investigate the expression pattern of *pMIR171A::H2B-GFP* and *pMIR171B::H2B-GFP* during AM (axillary meristem) initiation, *pMIR171A::H2B-GFP* line 14 and *pMIR171B::H2B-GFP* line 3 were grown in short days for 28 days and then in continuous light for 1–2 days. All these experiments were repeated with three biological replicates.

**Transactivation assay in tobacco.** To generate a *pMIR171A::LUC* reporter, the same *MIR171A* promoter described above was PCR amplified from Col-0 genomic DNA with the primers 5′-ACGGggtaccGATACTCCACTTTTAGGCTCCATCT T-3′ and 5′-ACGCgtcgacAAAGGGACTCTCTCATGCTTAAAG-3′ (restriction enzyme sites are in lower case) and cloned with KpnI and SalI into pGREEN800II-LUC[43].

To generate a *pMIR171A-mut::LUC* reporter, the *MIR171A-mut* promoter was PCR amplified with the primers 5′-ACGGggtaccGATACTCCACTTTTAGGCTC CATCTT-3′ and 5′-ACGCgtcgacAAAGGGACTCTCTCATGCTTAAAG-3′ and cloned with KpnI and SalI into pGREEN800II-LUC[43].

To generate a *pMIR171B::LUC* reporter, the *MIR171B* promoter was amplified with the primers 5′-ACGGggtaccATATAAAACATGCTATTGCTTCTT-3′ and 5′- ACGCgtcga**C**TAAAACCACTCTTGTTCGACTATAATC-3′ and cloned with KpnI and SalI into pGREEN800II-LUC.

To generate a *pMIR171B::LUC::MIR171B 3′-terminator* reporter, the *MIR171B* promoter was amplified with the primers 5′-ACGGggtaccATATAAAACATGCT ATTGCTTCTT-3′ and 5′-ACAAgcggccgcTAAAACCACTCTTGTTCGACTAT AATC-3′ and cloned with KpnI and NotI into pGREEN800II-LUC to generate *pMIR171B::LUC*. Then the *MIR171B* 3′ terminator was PCR amplified from Col-0 genomic DNA with primers 5′-CTAGtctagaAAGATAGTTATTATAACCTTA AAG-3′ and 5′- CTAGtctagaGAGTCCTTATTGTTGTGCCTTTTTA-3′ and

cloned with XbaI into *pMIR171B::LUC* to get *pMIR171B::LUC::MIR171B 3′-terminator*.

The effector ATML1 (35S::ATML1-GFP) was generated by introducing ATML1 cDNA into pMDC83. These dual-luciferase reporter constructs and indicated effectors (empty vector or ATML1) were introduced into *N. benthamiana* leaves through *Agrobacterium* infiltration. The activities of firefly luciferase (LUC) and Renilla luciferase (REN) were quantified 2 days after infiltration with a Dual Luciferase Assay kit (Promega), and luminescence was recorded using a 96-well Iuminometer (Tecan). The LUC activity was normalized to the REN activity (LUC/REN). The means and standard errors of LUC/REN were calculated from indicated numbers of biological replicates that are described in the figure legends. Each set of experiment has been independently repeated at least two times.

**Yeast one hybrid.** To generate the reporters of *pMIR171A::lacZ*, *pMIR171A-mut:: lacZ*, and *pMIR171B::lacZ*, the promoter of *MIR171A*, the mutated promoter of *MIR171A* (named as *MIR171A-mut*) and the promoter of *MIR171B* was subcloned, respectively with KpnI and SalI into pLacZi (Clontech).

To generate the *pMIR171A-F1::lacZ* reporter, the fragment 1 (F1) of the *MIR171A* promoter was amplified with the primers 5′- ACGGggtaccATGTGAA TGAAGCCAAGAATCA-3′ and 5′- ACGCgtcgacTTTCTTTTCAGGATTACT GTTTTCG-3′ and cloned with KpnI and SalI into pLacZi. To generate the *pMIR171A-F2::lacZ* reporter, the fragment 2 (F2) of the *MIR171A* promoter was amplified with the primers 5′-ACGGggtaccGGGTTCATAACTTGCTTGAGG-3′ and 5′-ACGCgtcgacTTTCCTTCTTGAGGAGGTTCA-3′ and cloned with KpnI and SalI into pLacZi. To generate the *pMIR171A-F3::lacZ* reporter, the fragment 3 (F3) of the *MIR171A* promoter was amplified with the primers 5′- ACGGggtacc CCTCCTCAAGAAGGAAAGACAG-3′ and 5′- ACGCgtcgacAGTGAGAATGTC TGTGGGGAGT−3′ and cloned with KpnI and SalI into pLacZi.

To generate the *pMIR171B-F1::lacZ* reporter, the fragment 1 (F1) of the *MIR171B* promoter was amplified with the primers 5′-ACGGggtaccATATAAAA CATGCTATTGCTTCTT-3′ and 5′-ACGCgtcgacCAAGTACCCAAAAACACT GAAAAA-3′ and cloned with KpnI and SalI into pLacZi. To generate the *pMIR171B-F2::lacZ* reporter, the fragment 2 (F2) of the *MIR171B* promoter was amplified with the primers 5′-ACGGggtaccTGCAAAGTCAATTATTTCTT TAAGC-3′ and 5′-ACGCgtcgacTCGGCCGCTAGCAAAGTAT-3′ and cloned with KpnI and SalI into pLacZi. To generate the *pMIR171B-F3::lacZ* reporter, the fragment 3 (F3) of the *MIR171B* promoter was amplified with the primers 5′- ACGGggtaccATGTTGATTACCCATATACTTTGCT-3′ and 5′-ACGCgtcgacTC CCACGAGCTTTCAATTTAGT-3′ and cloned with KpnI and SalI into pLacZi. To generate the *pMIR171B-F4::lacZ* reporter, the fragment 4 (F4) of the *MIR171B* promoter was amplified with the primers 5′-ACGGggtaccCCTGATAATTTGA TATCATCGGTTG-3′ and 5′-ACGCgtcgacATGCGGCCAGTAGCAAAGT-3′ and cloned with KpnI and SalI into pLacZi. To generate the *pMIR171B-F5::lacZ* reporter, the fragment 5 (F5) of the *MIR171B* promoter was amplified with the primers 5′-ACGGggtaccCTTGCTTGCCTCTCTTCTTCA-3′ and 5′-ACGCgtcgac CCACTTAGCTCCAGAAAACCA-3′ and cloned with KpnI and SalI into pLacZi. To generate the *pMIR171B-F6::lacZ* reporter, the fragment 6 (F6) from the 3′ region of the *MIR171B* gene was amplified with the primers 5′-ACGGggtaccTTTC ATCAAGTCGGTCCACA-3′ and 5′-ACGCgtcgacATCGATCGCATCTTTGG ATT-3′ and cloned with KpnI and SalI into pLacZi.

To generate the *pMIR171C::lacZ* reporter, the promoter of *MIR171C* was amplified with the primers 5′-ACGGggtaccTAATTCAAACCGAATTAGACC AAAA-3′ and 5′-ACGCgtcgacTCGACTCTTCAGTTGCTTATTACACCA-3′ and cloned with KpnI and SalI into pLacZi. To generate the *pMIR171C-F1::lacZ* reporter, the fragment 1 (F1) of the *MIR171C* promoter was amplified with the primers 5′-ACGGggtaccCATGCACTATTCATAGACCCATGT-3′ and 5′-ACG CgtcgacTCATGCATAAGCTTGCTTGG-3′ and cloned with KpnI and SalI into pLacZi. To generate the *pMIR171C-F2::lacZ* reporter, the fragment 2 (F2) of the *MIR171C* promoter was amplified with the primers 5′- ACGGggtaccTTTCTTCAT CACCCTCTTCG-3′ and 5′- ACGCgtcgacTCGACTCTTCAGTTGCTTATTAC ACCA-3′ and cloned with KpnI and SalI into pLacZi. To generate the *pMIR171C-F3::lacZ* reporter, the fragment 3 (F3) from the 3′ region of the *MIR171C* gene was amplified with the primers 5′- ACGGggtaccCCATGAACTAACCAGACGATCA-3′ and 5′- ACGCgtcgacTGTAATTAAGATCTGCGACCACTTC-3′ and cloned with KpnI and SalI into pLacZi.

To generate the *pMIR170::lacZ* reporter, the promoter of *MIR170* was amplified with the primers 5′-GCCCaagcttTCCTAACCTGTTCCTTTTGA-3′ and 5′-ACGC gtcgacGAAACATAGTGTATCTGTTTTGTAA-3′ and cloned with HindIII and SalI into pLacZi. To generate the *pMIR170-F1::lacZ* reporter, the fragment 1 (F1) of *MIR170* promoter was amplified with the primers 5′-GCCCaagcttGGTGGATTT CAAGGGTATGG-3′ and 5′-ACGCgtcgacAACCGTCCCTCTTTCGTTTT-3′ and cloned with HindIII and SalI into pLacZi. To generate the *pMIR170-F2::lacZ* reporter, the fragment 2 (F2) of the *MIR170* promoter was amplified with the primers 5′-GCCCaagcttACGAAAGAGGGACGGTTACA-3′ and 5′-ACGCgtc gacACTGTAAGGAGATTAAGAAGAAGG-3′ and cloned with HindIII and SalI into pLacZi. To generate the *pMIR170-F3::lacZ* reporter, the fragment 3 (F3) from the 3′ region of the *MIR170* gene was amplified with the primers 5′- GC CCaagcttATCGGATGCTCCTTTCTCCT-3′ and 5′- ACGCgtcgacCCAAGAATG GCCTTCCTACA-3′ and cloned with HindIII and SalI into pLacZi.

To generate the reporter strains, each pLacZi construct described above was transformed and integrated into the yeast YM4271 (Clontech). *ATML1* and *PDF2* cDNAs were cloned into pENTR-D-TOPO gateway entry vector (invitrogen) and then cloned into pDEST22 through LR reaction. These effector constructs were then transformed into each yeast reporter strain. The β-galactosidase (β-gal) activity was quantified using ONPG as substrate and in 96-well plate reader (Tescan) as previously described[44]. The fold of induction is calculated based on the β-gal activities for each transcription factor normalized to that from the empty vector controls. The means and standard errors of relative β-gal activities were calculated from indicated numbers of biological replicates that are described in the figure legends. Regarding the fold of induction in all the quantitative assays in Y1H, the cutoff line was set as two and any value less than two was not considered as positive interaction.

The x-gal lifting assay was performed as previous described[45]. Each set of experiment has been independently repeated two times.

**Computational modeling and simulation**. The current computational model is simulated on the same dome-shaped template (1216 overlapping spheres to mimic the SAM of Arabidopsis)[9]. The cell positions, cell sizes, and cell neighbors are defined in the 3D template[9]. Neighboring cells were identified by searching for the overlaps among spheres. Similar multiple spheres based 3D templates have been developed and used in previous modeling research on the transcriptional circuits and gene expression patterns involving *WUS*, *CLV3*, *HAM*, and/or hormone cytokinin in the SAM[8,46–48].

Since this model focuses on the gene expression patterns in the SAM and there is no clear experimental evidence for fluxes of the gene products (*ATML1*, *miR171*, and *HAM*) into and/or out of Arabidopsis SAMs, the boundary condition for this current model is set to be no flux boundary[9].

Dynamics of three key native molecules, namely microRNA171 ([*miR171*]), *HAM mRNA* ([*HAMr*]), and HAM protein ([*HAMp*]) are described using the system of differential Eqs. (1)–(3). To simulate the H2B-GFP reporter for *MIR171* transcriptional activity, we added two more equations for simulation of the patterns of the *pMIR171* reporter, which can be compared with and validated by our experimental data from confocal imaging of this reporter line. Since H2B-GFP only function as a reporter, there is no feedback regulation from the H2B-GFP reporter to the other components in the system described in Eqs. (1)–(3). The equations are applicable to each of the cells in the template, and the cell indices are omitted in the equations as shown. Relations among different agents are illustrated in the schematic diagram (Supplementary Fig. 7).

$$\frac{d[miR171]}{dt} = k_{mirp}[ML1p] - k_{mirn}[miR171] + D_{mir171}\Delta[miR171] \tag{1}$$

$$\frac{d[HAMr]}{dt} = k_{hrp} - k_{hrnm}[HAMr][miR171] - k_{hrnh}[HAMr] \tag{2}$$

$$\frac{d[HAMp]}{dt} = k_{hpp}[HAMr] - k_{hpn}[HAMp] \tag{3}$$

$$\frac{d[GFPr]}{dt} = k_{grp}[ML1p] - k_{grn}[GFPr] \tag{4}$$

$$\frac{d[GFPp]}{dt} = k_{gpp}[GFPr] - k_{gpn}[GFPp] \tag{5}$$

Equation (1) describes dynamics of miR171. $k_{mirp}$ is the parameter for miR171 production. [ML1p] is the concentration of functional ATML1 and its homolog PDF2 proteins. We assumed that miR171 production is activated by ATML1 and its homolog. This assumption is supported by four lines of evidence from our experimental results in this study: (1). ATML1 protein directly binds to the *MIR171* promoters; (2). The *MIR171* transcriptional reporter is induced by ATML1 in tobacco cells.; (3). The expression of the *MIR171* transcriptional reporter in epidermis is lost in *atml1 pdf2* double mutant plants; (4). Mutations in the ATML1 binding sites in the *MIR171A* promoter result in the abolishment of activation of *MIR171* transcriptional reporter both in the transient activation assays and in the SAM. $k_{mirn}$ is the parameter for miR171 degradation, and we assumed a constant rate for miR171 degradation throughout the SAM. In the third term of Eq. (1), we assumed that the epidermis produced miR171 RNA will move into deeper layers with a low diffusion constant, leading to a high to low gradient of miR171 RNA from the epidermal layer to deeper layers in the SAM. This assumption is supported by the results from the RNA in situ hybridization to probe the miR171 localization in the SAM (Supplementary Figs. 8 and 9). $D_{mir171}$ is the rescaled diffusion constant (rescaled by cell size) of miR171 among SAM cells. $\Delta$ is the discrete Laplace operator, and the contribution of $\Delta[miR171]$ to the derivative for miR171 RNA in cell i is given by $\sum_n([miR171]_n - [miR171]_i)$, where the sum is over neighboring cells. The set of neighbors is more restricted at boundaries.

Equation (2) describes the dynamics of the *HAM* mRNA. $k_{hrp}$ is the parameter for *HAM* mRNA production. Here we assumed that $k_{hrp}$ is constant in all cells within the SAM. This assumption is consistent with the experimental result that *pHAM2::H2B-GFP* reporter marker is highly expressed in all the layers of the SAM, including epidermis, sub-epidermis, and corpus. For degradation of *HAM* mRNA, we introduced both miR171 independent and dependent down-regulation terms. $k_{hrnh}$ is the parameter for the miR171 independent *HAM* mRNA degradation. We

assumed that *HAM* mRNA degradation is further directly activated by local miR171, and this model assumption is supported by previous work[11–13] and our current work. $k_{hrnm}$ is the parameter for the miR171 dependent *HAM* mRNA degradation.

Equation (3) describes dynamics of HAM protein. $k_{hpp}$ is the parameter for HAM protein production from *HAM* mRNA. $k_{hpn}$ is the parameter for HAM protein degradation.

Equation (4) describes dynamics of H2B-GFP mRNA. $k_{grp}$ is the parameter for *GFP* mRNA production activated by ATML1 protein. $k_{grn}$ is the parameter for *GFP* mRNA degradation.

Equation (5) describes dynamics of GFP protein. $k_{gpp}$ is the parameter for GFP protein production from *GFP* mRNA. $k_{gpn}$ is the parameter for GFP protein degradation.

In the model, the functional ATML1 and PDF2 proteins in the SAM is set as an input ML1p, and levels of *HAM* mRNA and HAM protein are outputs. In the wild type SAM, ML1p is only present in the epidermal layer, so the [ML1p] is set as 1 a. u. in the epidermal layer cells and 0 a.u. in all the other cell layers of the SAM during the entire simulation time. When ATML1 is ectopically activated, one set of [ML1p] is defined based on the experimental results from the *ATML1* RNA in situ hybridization in both wild type and the *35S::ATML1-GR* plant (Supplementary Fig. 12). In the L1 of the *35S::ATML1-GR* SAM, the level of *ATML1* looks slightly higher than that in the wild type. Therefore, [ML1p] in the L1 of the *35S::ATML1-GR* SAM was set to 1.1 a.u. At the deeper layers of the *35S::ATML1-GR* SAM, the level of *ATML1* is lower than that in the L1 layer of the same plant but higher than that in the deeper layers of the wild-type plant. Therefore, [ML1p] in deeper layers of the *35S::ATML1-GR* SAM was set to 0.4 a.u. This set of [ML1p] is not based on quantitative analyses, due to the qualitative or at most semiquantitative nature of the RNA in situ hybridization method. The 1.1 a.u./0.4 a.u. in the ATML1 overexpression line vs 1 a.u./0 a.u. in the wild type in the model are qualitatively comparable to the patterns shown from the in situ result.

To explore the effect of different [ML1p] values, a possible range of [ML1p] was assigned. Based on the RNA in situ images, the range is 1.05–1.5 a.u. for the L1 layer and 0.3–0.6 a.u. for the deep layers. Within this range, 24 different [ML1p] were randomly explored and used for the new simulations. Another possible set of [ML1p] values (1.3 a.u. in the L1 layer and 0.5 a.u. in deep layers) in this range is also included for the simulation. All these 25 sets of [ML1p] lead to suppressed *HAM* mRNA expression in deep layers of SAMs, which are largely comparable to the simulation result using the initial [ML1p] (1.1 a.u. in the L1 layer and 0.4 a.u. in deep layers), suggesting a robust response in the model when ATML1 protein is ectopically activated (Supplementary Movie 14).

In addition to the [ML1p] patterns (Supplementary Movie 14) indicated by the RNA in situ images, other possible [ML1p] patterns were explored, to examine whether differences between [ML1p] in L1 and that in deep layers are important for *HAM* mRNA expression when ATML1 is ectopically activated in SAMs. In summary, new simulations were carried out using 50 sets of [ML1p] from a different range of values: 1.05-1.5 a.u. for the L1 layer and 0.6-2 a.u. for the deep layers (Supplementary Movie 15). Within this range, 24 different sets of [ML1p] were randomly identified and used for the new simulations. Another possible set of [ML1p] (1.1 a.u. in the L1 layer and 0.7 a.u. in deep layers) in this range is also included for the simulation (Supplementary Movie 15). Moreover, 24 additional sets of [ML1p] with the identical values for all SAM layers were randomly explored in the range of 1.05–1.5 a.u. and used for the simulations (Supplementary Movie 15). One set of [ML1p] (with 1.1 a.u. in all layers of SAM) is also included for the simulation. All these 50 new sets of [ML1p] lead to suppressed simulated *HAM* mRNA expression in deep layers of SAMs (Supplementary Movie 15), which are largely comparable to the simulation result using the initial [ML1p] (1.1 a.u. in the L1 layer and 0.4 a.u. in deep layers), suggesting the repression of *HAM* mRNA expression in the SAM when ATML1 is ectopically activated is not sensitive to differences between [ML1p] in L1 and that in deep layers. Note that in simulations that compare the SAMs from wild-type plants and plants with ectopic activation of ATML1, the only difference was the [ML1p], and all the parameters (shown in Supplementary Table 1) were the same.

All simulations were carried out using the explicit forward Euler method with fine fixed time steps (0.01 h). At initial time point ($t = 0$ h), all five dynamic agents (miR171, *HAM* mRNA, HAM protein, H2B-GFP mRNA from *MIR171* transcriptional reporter, H2B-GFP protein from *MIR171* transcriptional reporter) were set as 0 a.u. All simulations were carried out from initial time point to steady stages. To visualize the gene expression patterns in deeper layers in the simulation results, only half of the simulated SAM was shown in figures.

In the five differential equations in the model, Eqs. (1) and (2) are the two equations that determine the dynamics of miR171 and *HAM* mRNA expression patterns, which are central components of our model. Equation (3) simply describes the universal relation between HAM protein and *HAM* mRNA, which does not provide novel property for *HAM* gene expression pattern. Equations (4) and (5) describe the dynamics of the H2B-GFP mRNA and H2B-GFP protein of the *MIR171* transcriptional reporter reflecting the regulation of the *MIR171* promoter, which do not feedback regulate miR171 or *HAM* mRNA levels, thus not essential for understanding the behavior of model in terms of miR171 and *HAM* mRNA pattern formation. Due to the above-mentioned reasons, the six parameters involved in Eqs. (1) and (2) were the focus for further exploration and analysis. The selected parameter values ranging from 40 to 200% of the initial value

(Supplementary Table 1) were explored and sampled at 2% intervals. In total, from simulated results using 481 sets of parameters, 235 sets of parameters can lead to the apical-basal patterns of *HAM* mRNA in the SAM qualitatively comparable to experimental observations. See Supplementary Table 2 and Supplementary Data 1 for the ranges and values of these parameters.

In addition to the local search based on the initial set of parameters (Supplementary Table 2, Supplementary Data 1), an unbiased random search was performed using quasi Monte Carlo method. In summary, a total of 173 new sets of different parameters were identified (Supplementary Data 3) from 20,000 sets of random parameters (Supplementary Data 2). Simulation using each of the 173 parameter sets is able to qualitatively represent the *HAM* mRNA pattern experimentally observed in the wild type SAM (Supplementary Movie 13). Details of the random search are listed below.

First, the low and high limit of each parameter was defined for the random search, based on knowledge of biological context in general and specific features of the regulatory system in the SAM being studied. Because a wild type SAM is dynamically stable, the production and degradation of the same agent (RNA/protein) need to be balanced. Therefore, the production and degradation rates for the agent were set to vary within a 100-fold range. In addition, because the miR171 regulates the *HAM* mRNA levels through direct binding and cleavage, the difference of turnover rates for these two agents is set within a 100-fold range. With these considerations, the search for parameter values was performed in the range of 0.1-1 for miR171 production rate, miR171 degradation rate, *HAM* mRNA production rate, miR171 dependent *HAM* mRNA degradation rate, and miR171 independent *HAM* mRNA degradation rate (the unit for each parameter is listed in Supplementary Table 1). Further, because movement of microRNAs between cells in the SAM is generally limited[33], the search for parameter values of miR171 rescaled diffusion constant (rescaled by cell size) was defined in the range of 0.001–0.1 (area a.u./h).

Within the defined ranges, Quasi Monte Carlo Method was used to randomly sample 20,000 sets of six different parameters uniformly on the log-scale (Supplementary Data 2). Then, simulations were performed using each of these 20,000 random parameter sets in the 3D template to search for more solutions for *HAM* mRNA patterns in the SAM. After comparing these simulation results (Supplementary Movie 13), 173 new sets of solutions (shown in Supplementary Data 3) were identified, which are different from the initial set of parameters (shown in Supplementary Table 1).

The sensitivity of the total *HAM* mRNA (model output) to variations of six key parameters was analyzed[47]. Specifically, the parameters are increased by 1%. The sensitivity is computed as: $\frac{\Delta m}{\Delta p} \times \frac{p}{m}$. p is the original parameter value. $\Delta p$ is the variation of the parameter. m is the total *HAM* mRNA expression. $\Delta m$ is the variation of the total *HAM* mRNA expression. This sensitivity of all 6 key parameters is computed using 235 sets of parameter values (Supplementary Fig. 10) or 173 sets of parameters defined by the random search (Supplementary Fig. 11). In general, the model output shows robust behavior to variations of the six parameters. The total level of *HAM* mRNA shows higher degree of sensitivity to three parameters, including *HAM* mRNA production rate ($k_{hrp}$) and miR171 independent *HAM* mRNA degradation rate ($k_{hrnh}$), which directly control the production and degradation of *HAM* mRNA, respectively, and ($k_{mirn}$), which determines the degradation of miR171.

**RT-qPCR**. The levels of gene expression in both Mock-treated and Dex-treated *35S::ATML1-GR* plants were quantified through the real-time RT-PCR. *35S:: ATML1-GR* plants lines 5 and 9 were grown in short days for 20 days. After that, they were sprayed with mock (no Dex) or with 10 μM DEX solution. After 24 h, the whole shoots were cut off from the top of hypocotyls and harvested for RNA extraction and RNA was isolated using RNeasy Mini Kit (Qiagen). Reverse transcription was performed from 1.2 μg total RNA using SuperScript Reverse Transcriptase (ThermoFisher). Quantitative PCR was performed using SYBR Green qPCR Master Mix (Bimake) on a BioRad CFX96. The primers for *UBC* (as the internal control) qPCR are described previously[7]. The primers for *GFP* are 5′-GA ACCGGCATCGAGCTGAA-3′ and 5′-TGCTTGTCGGCCATGATATAG-3′. The primers for *MIR171A*: 5′-GATATTGGCCTGGTTCACTC-3′ and 5′-CCACAAA GTCCAAAATAGAG-3′ and for *MIR171B*: 5′-GGAGCTAAGTGGAGATTAT AG-3′ and 5′-GGTTATAATAACTATCTTTGCC-3′ as described previously[13]. The primers of *HAM1* and *HAM2* qPCR were designed to span their *miR171* targeting sites. For *HAM1*: 5′-AAACAACAACGGCGACCA-3′ and 5′-CTTT GAAACGGAGACTTGTGG-3′ were used. For *HAM2*: 5′-CAAACGGCGG AGATAACAAT-3′ and 5′-CTGTGGAACGGAGGTTTAGG-3′ were used. The mean and standard error was calculated from three independent biological replicates.

**Reporting summary**. Further information on research design is available in the Nature Research Reporting Summary linked to this article.

## Data availability
All data are available from the corresponding author upon request. The source data underlying Figs. 2b–l, 3b–i, 4b, c, e, f, 5a–c, 7i–l, and Supplementary Figs. 2, 1, 4, 10, 11, and 17 are provided as a Source Data file.

## Code availability
The code is available from the corresponding author upon request.

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

## Acknowledgements

We thank the Purdue Bindley Bioscience Imaging Facility for access to the ZEISS LSM880 Laser scanning confocal microscope. We appreciate the generous support and help from Dr. Elliot Meyerowitz. We also thank Dr. Eric Mjolsness for the helpful discussions and suggestions on the parameter tests in the computational model. The work was supported by Purdue University start-up package and funds from Purdue Center for Plant Biology to Y.Z.

## Author contributions

H.H., X.L., and Y.Z. conceived the research direction, H.H., L.L., X.L., and Y.Z. performed the experiments, A.Y. performed the model simulation, A.Y. and B.F. analyzed the sensitivity of model parameters, Yf.Z. contributed reagents and commented on the results, H.H., A.Y., X.L., and Y.Z. wrote the manuscript. All the authors read and agreed to the manuscript.

## Competing interests

The authors declare no competing interests.
