## [Peer Review File · Nature Communications]

Reviewers' comments:

Reviewer #1 (Remarks to the Author):

Manuscript: The apical-basal patterning of Arabidopsis shoot apical meristems is defined by a signal cascade initiated at the epidermis

In this manuscript, the authors address the regulation of the expression pattern of the HAIRY MERISTEM 1 gene in the basal shoot apical meristem of Arabidopsis thaliana. The manuscript starts with the observation that the four homologous miR171/170 genes, whose products target HAM transcripts, are specifically transcribed in the L1 layer of the SAM. Recognizing that this domain strongly resembles the expression pattern of the transcription co-factors ATML1 and PDF2, and that MIR171 promoters contain ATML1/PDF2 binding motifs, Han et al. provide evidence that ATML1 and PDF2 indeed regulate expression of those genes.

General comments

The paper is well written and has very nice videos and figures. The topic is of high interest to understand meristem development. There are however several limitations that decrease the novelty and the significance of the paper.

1) The observation that miR171A is expressed exclusively in the L1 layer, is able to move to the inner layers and confines the HAM expression to the inner cells was published before in Takanashi et al., 2018.

2) The author's model is based on that mi171 moves from the L1 into deeper layer. However, although this is a cornerstone of the patterning, the author's did not show miR171 localisation, although the in situ hybridization technique for small RNAs is well established.

3) I do not see a benefit of mathematical modelling in this specific case. There is one input, one output and a simple linear relationship: ATML activates miR171 which represses HAM. Instead of the elaborate simulations one could simply say, we tested the hypothesis that mir171/HAM are downstream of ATML by ectopic expression of ATML1 under an inducible 35S promoter system.

Minor comments:

Takada & Jürgens (2007) observed that the L1 box of pATML requires a nearby functional WUS box to establish the full L1 pattern. Is this also true for pmiR171?

Could the authors comment on whether this pathway is also active in other tissues, roots in particular? HAM as well as ATML1 and miR171 are expressed in roots.

In the ectopic ATML1 expression lines, ATML1 is expressed under an inducible constitutive promoter. Fig 5h, S6, S7 and S8h don't look like MIR171 is expressed in all cells where ATML1 is expressed. How do the authors explain that?

Page 4 from line 5

The canonic L1 motif sequence published by Abe et al. (2001) is TAAATG(C/T)A. The authors use altered motifs in their work (TTAAATGC for miR171A and TAAATG for miR171B). Could the authors please explain their reasoning for this? If I use these truncated motifs as a template, I find three motifs for pmiR171a in the Araport11 genome sequence (-1122, -618, -290 from transcription start). If we're also taking the Minus strand into account, there are also three in pmiR171b (-1218, -1070, -208 from transcription start). What about the promoters of miR170 and miR171c? How to explain their weak expression?

Page 5 line 3

How many independent lines were observed?

For the discussion

Do atml1 pdf2 double mutants exhibit HAM overexpression or HAM ectopic expression phenotypes?

Figure 5 o,p legend

Please give the number of observations made

Figure 5 a-d, m,n; Figure 6 e-l, u-x

Page 2 line 3:
should read "HAIRY"

Page 2 line 22:

What is meant by miR171 "can be" expressed in a wide range of tissues? Is it expressed there or not?

Page 3 line 4:
should read "fluorescent"

Page 3 line 7:
should probably read "are restricted to the epidermis"

Page 3 line 12:
should probably read "expressed in the epidermis"

Page 8 line 20

There is a space missing between the period and "We"

Figure 1 caption line 8 and line 12

Instead of "responsible" maybe "sensitive" or "responsive"?

Reviewer #2 (Remarks to the Author):

This manuscript by Han and colleagues describes how factors specifying the identity of the epidermis trigger the production of miR171 from the L1 in the shoot apical meristem that in turn specify the pattern of HAM genes. To do so the authors use a combination of live-imaging, genetics and computational modeling. While several recent papers have described the role of HAM proteins in controlling the location and the activity of the stem cell niche in the meristem, this work allows to understand how the distribution of HAM proteins might be established and provides a much deeper understanding of how interactions between layers act in the control of meristem function.

However I have some concerns with some of the data and first with the model. Page 5, the authors claim that they have developed "a new 3D computational model". This model is largely based on previous work by the lead author (Ref 9 of manuscript); so it is not really new and this should be corrected (also on the middle of Page 9) . The authors should also detail in the main text and main figure what the model is.

Also Page 6, there are 2 paragraphs first discussing the model and presenting in silico experiments ("We further tested the hypothesis...") then performing wet experiments ("We also reproduced ..."). The presentation of the first paragraph (modeling) is very perturbing because the authors present the design of the wet experiment and rather than describing what they do in the model. They need to describe this from the point view of the model (I.e. what do they modify in the model e.g. forcing the L1 input in all layers) to avoid confusion before presenting the design of the wet experiment (the use of 35S::ATML1 etc).

Also all simulations are done with fixed parameters. So this makes it difficult to judge what is the robustness of the model or whether it works only with this set of parameters that are not justified whatsoever. The use of the model in the manuscript is also purely qualitative (and it should be mentioned if it stays like that) while the authors are in capacity to acquire quantitative data that could feed the models. E.g. Page 3 and Figure 1, concerning the gradients of HAMs and notably the one of HAM2 it would be interesting to have a quantification of these. Can the models really reproduce quantitatively these gradients? This is an important point as the authors claim throughout the manuscript that the simulation and wet experiment results are "the same" or "almost the same". At this stage it is difficult to judge the solidity of these conclusions.

Also some modeling data presented in the manuscript are difficult to understand. Fig 5 a-d: either

I have missed something or what is done in the model is forcing the L1 input in all layers. And here the "Dex induction" (see my comment above: this needs to be changed) changes the values in L1 vs internal layers but has a mild effect on internal tissues. This is likely because the "induction" is mild and just additive in comparison to mock but this requires explanations. This is the case also for Fig S9: why are the 2 "mocks" and "Dex" different? The authors really need to significantly expand their description of the modeling work in the manuscript.

From the modeling work, one key expected outcome of the expression of ATML1 throughout the SAM is the induction of the MIR171 genes in all layers. This is nicely shown for MIR171B in Fig5 but for MIR171A Fig S8 raises quite a lot of question. The authors are surprisingly showing a zoom on one of the flower rather than showing the SAM and the induction seems far less efficient and patchy. The authors need to show what happens in the SAM. Rather than claiming that the data perfectly support their model (Page 6 concluding phrase, last paragraphe), they should discuss the fact that the regulation might be more complex than what they claim (which is not a problem but a biological reality).

Page 8 on de novo axillary stem cell niches, in the concluding paragraph the authors give the feeling that they have been able to confront the dynamical vision that they have in their simulation with the actual dynamics of the patterning of the lateral meristem. While this could be true for the establishment of the HAM gradient, the wet data do not allow to judge when mir171 is turned on in relation to ATML1 expression. The authors need to carefully check this entire section and its conclusion for accuracy.

Other comments:

- P2 top of the page: "HARIY MERISTEM" should be "HAIRY MERISTEM"
- Page 6 middle of the page: "ATML1 directly activates 171 in the SAM"; 171 should probably be miR171.
- End of Page 6: "... Dex treatment compared with mock control (Fig. 5)." Please specify the panel(s) on Fig 5 that is(are) concerned.
- Page 8 1st paragraph of Discussion: "MIR171A and MIR171B genes are the major contributor for miR171 activity" - this has not been demonstrated by the authors; either they need to provide data or this needs to be corrected.
- Page 10 last paragraph of discussion: "ATML1-miR171-HAM" - elsewhere this module is called L1-miR171-HAM which is probably more accurate as the authors suggest a role for PDF2. This needs to be corrected and the authors need to ensure that they use the same name throughout the manuscript.

Reviewer #3 (Remarks to the Author):

The authors study the role of microRNA within the apical-basal patterning in the SAM. This is an interesting problem well beyond the specific plant field. In this review I focus on the computational part of the manuscript and I leave the experimental part to others. I have a few major problems with this part:

1. The authors put the model quite prominently in the main text ("new 3D computational model"). However, they fail to sufficiently describe what they have done. Either they extend and clarify this part or it needs to be removed. It cannot be to the reader of a sound scientific paper to guess what was done. I suggest a cartoon like figure together with some, e.g., pseudo code in the MAIN text. The simulation details also need to be described somewhere and not in a previous paper.
2. The equations in the Methods section are insufficiently explained. Why is the activation of [HAMr] modelled in this way? What does the Delta mean in equation 1? The Laplacian operator? How is it discretised? What about boundary conditions?
3. I severely doubt the delay in equation 2 (degradation). I understand the intention, but I doubt that the implementation is correct. At least it is not consistent with the description the authors

give. It is NOT that it takes a while after complex formation until [HAMr] is removed from the system. It rather models a maturation time of [mir171], which may not what the authors want to model. A delayed removal would rather have a delay in both, [HAMr] and [mir171]. If the authors believe I am wrong they should provide for me and the interested reader a derivation of the delay in equation 2.

4. Where do the parameters come from? How certain are the authors about them and why? What about a sensitivity analysis? Usually these parameters are only defined in an uncertainty range. How does the model behave within the hypercube defined by all parameter ranges? A local analysis, based on parameter fiddling is not good enough here.

5. The relevant literature on modelling approaches of the SAM are not sufficiently mentioned nor cited.

As I said initially, I focus on the modelling part, as I have here severe doubts that this work lives up to the standard of Nature Communications. Nevertheless, the experimental part may do and a way out for the authors may be to remove the modelling part. Or to address my comments above.

Reviewers' comments:

Reviewer #1 (Remarks to the Author):

In this manuscript, the authors address the regulation of the expression pattern of the HAIRY MERISTEM 1 gene in the basal shoot apical meristem of *Arabidopsis thaliana*. The manuscript starts with the observation that the four homologous miR171/170 genes, whose products target HAM transcripts, are specifically transcribed in the L1 layer of the SAM. Recognizing that this domain strongly resembles the expression pattern of the transcription co-factors ATML1 and PDF2, and that MIR171 promoters contain ATML1/PDF2 binding motifs, Han et al. provide evidence that ATML1 and PDF2 indeed regulate expression of those genes.

General comments

The paper is well written and has very nice videos and figures. The topic is of high interest to understand meristem development. There are however several limitations that decrease the novelty and the significance of the paper.

1) The observation that miR171A is expressed exclusively in the L1 layer, is able to move to the inner layers and confines the HAM expression to the inner cells was published before in Takanashi et al., 2018.

Response: Thanks for the comment. Yes, we are aware of Takanashi et al., 2018 and we have cited it in our previous submission. Takanashi et al., 2018 reported that one microRNA, MIR171A, is specifically expressed in the epidermis of several different tissues during both embryonic and post-embryonic development in *Arabidopsis*, and their work suggested a role of the epidermis-expressed MIR171A in control of HAM1-mediated embryo development. The work we report here is largely beyond the observations reported in Takanashi et al., 2018, with greater breadth and depth. First, our work reveals the expression patterns of all four different MIR171/170 genes in the SAMs and SAM derived tissues, and we surprisingly found that the expressions of MIR171B, MIR171C and MIR170 are also restricted to the epidermis of the SAMs (Fig. 1). Following these observations, we then determined the molecular mechanism that defines MIR171 expression in the L1. Through extensive biochemical and molecular genetic analyses (Figs. 2-6), we defined the key transcription factors (ATML1 and PDF2) and the essential cis-regulatory elements that mediate the activation of MIR171 in epidermis. Most importantly, our work uncovers a link that is essential for the meristem development but is missing from previous studies. Previously, it appeared that ATML1 and HAM were totally unrelated, because ATML1/PDF2 proteins are specifically localized in the epidermis, whereas HAM proteins are excluded from epidermis. However, our work here shows that although ATML1/PDF2 do not directly regulate HAM, they regulate HAM through the mobile microRNAs, forming an L1-miR171-HAM signaling cascade. From a broader perspective, our work provides a model illustrating how transcription factors that are epidermis specific determine the apical-basal polarity of gene expressions in multiple layers of the SAM, which in our opinion is nontrivial and can be of great interest to fields of developmental biology, signaling transduction, transcriptional regulation, cell type determination, etc.

2) The author's model is based on that miR171 moves from the L1 into deeper layer. However, although this is a cornerstone of the patterning, the author's did not show miR171 localisation, although the in situ hybridization technique for small RNAs is well established.

Response: Thanks for the comments. Following the reviewer's suggestion, we have performed the RNA in situ hybridization of the miRNA171 using the published technique of the LNA probe (Javelle and Timmermans, Nature protocols 2012). Our results showed that the localization of endogenous miR171 is high in the L1 layer and low at the L2 and upper corpus (Supplementary Fig. 7), which have been confirmed by five independent biological replicates. This result is consistent with the conclusion that we previously made based on the comparison of the confocal images from the miR171 sensitive HAM2 reporter and the miR171 insensitive HAM2 reporter (Fig. 1 e, f). The localization of miR171 in the SAM is also consistent with the recent findings that in general, microRNAs can move in the SAM and the movement is highly regulated (Skopelitis et al., Nature communications 2019).

3) I do not see a benefit of mathematical modelling in this specific case. There is one input, one output and a simple linear relationship: ATML activates miR171 which represses HAM. Instead of the elaborate simulations one could simply say, we tested the hypothesis that mir171/HAM are downstream of ATML by ectopic expression of ATML1 under an inducible 35S promoter system.

Response: We apologize that we did not clearly describe the structure and importance of the computational model in the initial submission.

We partially agree with the reviewer, and if we consider the whole SAM as a single point, the relationship can be as simple as that more ATML1 expression leads to higher miR171 production and less HAM mRNA. However, the benefit of the model is to help resolve the complex spatial patterning of HAM expression in multiple cell layers in 3D, and the model includes 1216 individual cells at different positions in the SAM. Thus, our work here is to not only elucidate the linear relationship among ATML1, miR171, and HAM, but also illustrate the cell-cell communication of the signaling molecules and the spatial patterning of gene expression in the SAM. Our model simulation together with experimentation provide a comprehensive view on how the apical-basal polarity of *HAM* gene expression is established in multiple layers while the driving force of this process (ATML1) is restricted in the single epidermal layer of the SAM.

To help better understand the model, in the revised manuscript, we have included detailed descriptions in the main text, the methods, supplementary movies and the supplementary documents.

Minor comments:

Takada & Jürgens (2007) observed that the L1 box of pATML requires a nearby functional WUS box to establish the full L1 pattern. Is this also true for pmiR171?

Response: We thank the reviewer for this comment. We have carefully searched for the putative WUS binding site (TTAAT(G/C)(G/C)) (based on Lohmann et al. Cell 2001, and Takada & Jürgens Development (2007)) from the regulatory regions (promoters and 3' terminators) of the four MIR171/170 genes. We did not find any TTAAT(G/C)(G/C) sequence from the regulatory regions of MIR171A, MIR171C or MIR170. However, we found one (TTAATGC) motif from -317 and one (TTAATGG) motif from +178 in the MIR171B locus.

We would like to point out that, when we searched for the putative ATML1 binding sites, we found the TTAATGC (from -317 of MIR171B) site is part of the ATTTAATGC (Fig. 3a), which shares conserved DNA sequences with the previously reported the L1-box motif and HD-ZIP IV binding sites (GCATT(A/T)AATGC identified in Nakamura et. al the Plant Journal 2006). We tested this putative cis-element in the DNA fragment (F4) in our Y1H and EMSA assays, and we found that it interacts with ATML1/PDF2 (Fig 3a, c-d, g-h).

The other putative WUS box (TTAATGG) we identified (from +178 at the 3' region of the MIR171B gene) is ~500bp away from a putative L1 box from the 3' region of the MIR171B gene (Supplementary Fig. 4).

Taken together, our findings suggest that the conclusion in Takada & Jürgens Development (2007) is likely conserved for MIR171B. The functional significance of putative WUS boxes in control of MIR171B expression still needs further studies in the future.

Could the authors comment on whether this pathway is also active in other tissues, roots in particular? HAM as well as ATML1 and miR171 are expressed in roots.

Response: Thanks for raising a very interesting question. We have performed additional confocal imaging of the reporters, and we found that the expression of ATML1 and miR171 are separated in roots (Supplementary Fig. 14) (Supplementary Movies 13, 14). ATML1 is expressed in the epidermis of root apical meristems (Sessions et al., the Plant Journal 1999). In contrast, all four MIR171/170 reporters do not have detectable expression in the root apical meristem. Among the four MIR171/MIR170 genes, MIR171B and MIR171C are specifically expressed in one cell layer inside the differentiation zone of the root (Supplementary Fig. 14) (Supplementary Movies 13, 14). These results suggest that the L1-miR171-HAM pathway is not active in roots.

We also examined a few other tissues in shoots. Our imaging results showed that MIR171A and MIR171B are specifically expressed in the flower meristems and young leaves (Fig. 1 a-b). Their expressions are also highly induced in flower meristems and leaf primordia upon the Dex treatment in the 35S::ATML1-GR plants (Fig. 7h, i-k, Supplementary Figs. 8-11) (Please note that the qPCR experiment for MIR171A and MIR171B (Fig. 7i-k) also only involved the above-ground tissues, which was described in the methods from the initial submission). These results suggest that the pathway is active in the shoot tissues we analyzed.

Based on these results, we conclude that the L1-miR171-HAM signaling cascade that we proposed and validated in this study is highly specific to the SAM and the SAM derived young tissues (leaf primordia and flower primordia, for example). We hypothesize that either ATML1 needs additional transcriptional cofactors to turn on miR171 in the root, or there are key transcriptional repressors to keep miR171 off in the root epidermis. It will be very interesting to

examine the specific regulation of MIR171B and MIR171C in the roots in the future. We have included this discussion in the main text.

In the ectopic ATML1 expression lines, ATML1 is expressed under an inducible constitutive promoter. Fig 5h, S6, S7 and S8h don't look like MIR171 is expressed in all cells where ATML1 is expressed. How do the authors explain that?

Response: Thanks for pointing out this concern. Since the reviewer 2 also had similar concerns and provided helpful suggestions, we would like to respond here to both reviewers. We have examined the expression pattern of ATML1-GR in our 35S::ATML1-GR/pMIR171B::H2B-GFP line by RNA in situ hybridization to the mRNA of ATML1 (Supplementary Fig. 10), and we included the SAM of the wild type as a control. Apparently, although the 35S promoter is generally thought to be a constitutive promoter, ATML1-GR driven by the 35S promoter is not ubiquitously expressed in every cell in the SAMs of 35S::ATML1-GR/pMIR171B::H2B-GFP (Supplementary Fig. 10). Therefore, we think the un-even/patchy signal observed in Fig. 5h, S6 and S7 (now the Fig. 7h, Supplementary Fig. 8 and 9 in the revised manuscript) is due to this nature of the 35S promoter. In our opinion, comparing the patterns of ATML1 and MIR171B (Fig.7h, Supplementary Figs. 8-10) further supports our conclusion that ATML1 can directly and quickly turn on MIR171B in the SAMs and primordia.

To carefully examine the induction of MIR171A, we have performed a new experiment (Supplementary Fig. 11). We grew the Mock and the Dex treated samples in the same condition and fixed them at the same time. We then performed RNA in situ hybridization on these samples using the identical procedure. Our new results showed that indeed MIR171A was greatly induced in the shoot apex one day after the Dex treatment in the 35S::ATML1-GR plants, which is consistent with the qPCR results (Fig. 7j in the revised manuscript, also included in our previous submission). However, slightly different from the MIR171B, we noticed that in the SAM, the induction of MIR171A is not as efficient as that of 171B upon activation of 35S::ATML1-GR (with strong activation in young leaves but only mild and uneven induction in the SAM), especially at the deep cell layers of the SAM. These results suggest potential differences in the regulation of the MIR171A promoter vs. the MIR171B promoter in distinct cell types in the SAMs, which is also interesting. We have included this discussion in the main text.

Page 4 from line 5

The canonic L1 motif sequence published by Abe et al. (2001) is TAAATG(C/T)A. The authors use altered motifs in their work (TTAAATGC for miR171A and TAAATG for miR171B). Could the authors please explain their reasoning for this? If I use these truncated motifs as a template, I find three motifs for pmiR171a in the Araport11 genome sequence (-1122, -618, -290 from transcription start). If we're also taking the Minus strand into account, there are also three in pmiR171b (-1218, -1070, -208 from transcription start). What about the promoters of miR170 and miR171c? How to explain their weak expression?

Response: We sincerely appreciate the comments from the reviewer. These comments promoted us to perform a series of new experiments, and the new results have helped improve our manuscript. Shortly speaking, these results have validated our previous conclusion and provided new information. Please see details below.

Initially, when we looked for the putative binding sites for ATML1/PDF2 from the promoters of MIR171A and MIR171B, we incorporated the information from two research papers: the TAAATG(C/T)A sequence identified by Abe et al., the Plant Journal 2001, and the GCATT(A/T)AATGC consensus sequence for ATML1 and PDF2 identified by (Nakamura et al., the Plant Journal 2006). Since the sequences reported in two papers are largely overlapping but slightly different, we searched the sequence in common, TAAATGC or TAAATG, from the forward direction of the promoters of MIR171A and MIR171B as the putative minimal binding sites for ATML1/PDF2, and we then tested the DNA-protein interactions using independent approaches. These experimental results are presented in the initial submission and in the revised manuscript.

Following the helpful suggestions and comments from the reviewer, and in order to comprehensively identify ATML1 binding sites, we performed new searches in all four MIR171/MIR170 genes in both forward and reverse directions, using different versions of the putative L1 motif, containing either TT(A/T)AATG(C/T), T(A/T)AATG(C/T), TAAATG(C/T)A or even the shorter version with conserved TAAATG. We performed Y1H and EMSA using DNA fragments containing these putative canonical L1 motifs and the new results are shown in Figs. 2-4.

Our results demonstrated that the consensus sites from MIR171 promoters interact with ATML1 /PDF2, consistent with the L1 box motifs identified in the previous studies (Abe et al., the Plant Journal 2001)(Nakamura et. al the Plant Journal 2006). We further demonstrated that in the MIR171A promoter, two sites including the TTAAATGC sequence (in MIR171A-F1 and MIR171A-F3) and one site including the ATTAATGT sequence (in MIR171A-F2) strongly interacts with ATML1/PDF2; one site from the MIR171B promoter including the ATTTAATGC sequence (in MIR171B-F4) also strongly interacts with ATML1/PDF2. Thus, we conclude that that the (A)TT(A/T)AATG(C/T) sequences from the promoters of MIR171A/B genes mediate the ATML1 binding. In addition, DNA sequences from these promoters containing TAAATG may also function as the ATML1 binding sites, depending on the context or the flanking sequences of each core element. We have included these results and discussions in the main text.

As shown in Fig. 4, the promoters of MIR171C and MIR170 also contain several L1 motifs and they interact with ATML1 in yeast. These new results suggest that the epidermis specific expression of these two genes may also due to the direct regulation by ATML1, and we have included this discussion in the main text. On the other hand, we hesitate to explain the low expression of MIR171C and MIR170 in the SAMs by simply relating it to the binding affinity between ATML1 and the promoter DNA. It is very common that genes directly regulated by the same transcription factor display highly variable expression levels, and multiple cis- and trans-regulatory components may contribute to the variation in the gene expression levels.

Page 5 line 3

How many independent lines were observed?

Response: 31 independent lines in total.

For the discussion

Do *atml1 pdf2* double mutants exhibit HAM overexpression or HAM ectopic expression phenotypes?

Response: ATML1 and PDF2 regulate a number of different targets and control the epidermal specification pathway. Based on the literature and our observation, the *atml1 pdf2* double mutants showed very severe defects that appear to be associated with the loss of the epidermal identity. The whole developmental program and cell identities of the *atml1 pdf2* double mutants also differ from that of the wild type. For these reasons, we do not think examining the HAM expression in *atml1 pdf2* double mutants will be informative. In addition, the phenotype of *atml1 pdf2* double is unlikely to be solely resulted from the mis-regulation of miR171, and thus, we hesitate to make direct comparison between the phenotypes of the *atml1 pdf2* double mutant and the HAM ectopic expression.

Figure 5 o,p legend

Please give the number of observations made

Figure 5 a-d, m,n; Figure 6 e-l, u-x

Response: In our previous manuscript, we have described the numbers of all the observations (the number of biological replicates) in details in the methods section. To help our readers better access this information, we have added them in the figure legend and main text in our revised manuscript.

Page 2 line 3:
should read "HAIRY"

Response: Thanks and we have made this revision.

Page 2 line 22:

What is meant by miR171 "can be" expressed in a wide range of tissues? Is it expressed there or not?

Response: Thanks and we have revised it in the main text. miR171 is expressed in a wide range of tissues as reported by several papers, and we have updated the reference in the main text.

Page 3 line 4:

should read "fluorescent"

Response: Thanks and we have made this revision.

Page 3 line 7:
should probably read “are restricted to the epidermis”
Response: Thanks and we have made this revision.

Page 3 line 12:
should probably read “expressed in the epidermis”
Response: Thanks and we have made this revision.

Page 8 line 20
There is a space missing between the period and “We”
Figure 1 caption line 8 and line 12
Instead of “responsible” maybe “sensitive” or “responsive”?
Response: Thanks and we have made these revisions.

Reviewer #2 (Remarks to the Author):

This manuscript by Han and colleagues describes how factors specifying the identity of the epidermis trigger the production of miR171 from the L1 in the shoot apical meristem that in turn specify the pattern of HAM genes. To do so the authors use a combination of live-imaging, genetics and computational modeling. While several recent papers have described the role of HAM proteins in controlling the location and the activity of the stem cell niche in the meristem, this work allows to understand how the distribution of HAM proteins might be established and provides a much deeper understanding of how interactions between layers act in the control of meristem function.

However I have some concerns with some of the data and first with the model. Page 5, the authors claim that they have developed "a new 3D computational model". This model is largely based on previous work by the lead author (Ref 9 of manuscript); so it is not really new and this should be corrected (also on the middle of Page 9) . The authors should also detail in the main text and main figure what the model is.

Response: Thank you so much for the comments. We apologize that we did not accurately describe the current model in our previous submission. In the current manuscript, we have revised the model description and avoided to claim it as a completely new model. We have also included more details describing the current model in the main text, methods, supplementary movies or other supplementary documents. Specifically, this model shared the SAM model template from our previous report (Zhou et al., Science 2018), but it addressed a different question with new equations and parameters. In the previous model, we used HAM concentration gradient as one key functional input (Zhou et al., Science 2018) to study the CLV3 gene expression patterns, which was the output. In this model, we used the epidermal specification factors (ATML1 and its close homolog PDF2) as the key input, and the functional

output include the concentration gradient of HAM in the SAM (ultimate functional output), the MIR171 promoter activity, and the patterns of miR171.

Also Page 6, there are 2 paragraphs first discussing the model and presenting in silico experiments ("We further tested the hypothesis...") then performing wet experiments ("We also reproduced ..."). The presentation of the first paragraph (modeling) is very perturbing because the authors present the design of the wet experiment and rather than describing what they do in the model. They need to describe this from the point view of the model (I.e. what do they modify in the model e.g. forcing the L1 input in all layers) to avoid confusion before presenting the design of the wet experiment (the use of 35S::ATML1 etc).

Response: Thanks for the suggestion. We have extensively revised the model description in the main text. In the revised manuscript, we first described step by step on how to establish this model to simulate miR171 and HAM in a wild type SAM. We then described the modification of the input when the activity of ATML1 is turned on and the values of the input in different cells. After that, we start to describe on how we performed experiments using the Dex inducible system to turn on ATML1 activity in vivo. With the reviewer's suggestion, we believe we have improved the presentation of our work.

Also all simulations are done with fixed parameters. So this makes it difficult to judge what is the robustness of the model or whether it works only with this set of parameters that are not justified whatsoever.

Response: Thanks for this critical comment. Since the reviewer 3 had a similar comment, we would like to respond here to both reviewers. We totally agree with the reviewers and we think that it is very important to work out a number of solutions with different parameter values to vigorously test the model. Thus, we performed a large amount of additional computational tests and we have included the results in the revised manuscript.

Specifically, our model contains six key parameters that control the dynamics of miR171 and the HAM mRNA expression pattern (Supplementary Tables 1 and 2). We have explored the working ranges of these six key parameters (Supplementary Table 2). In total, we defined 235 sets of different combinations of parameter values (Supplementary Table 3). Using each set of parameter values, we can simulate patterns of HAM mRNA that are qualitatively comparable to the observed pattern in the WT SAM (shown as Supplementary Movies 7-12). Please note that we have performed all the simulation work with different parameter values in the 3D template, and importantly, these tests demonstrate that the system we established is robust.

The use of the model in the manuscript is also purely qualitative (and it should be mentioned if it stays like that) while the authors are in capacity to acquire quantitative data that could feed the models. E.g. Page 3 and Figure 1, concerning the gradients of HAMs and notably the one of HAM2 it would be interesting to have a quantification of these. Can the models really reproduce quantitatively these gradients? This is an important point as the authors claim throughout the manuscript that the simulation and wet experiment results are "the same" or "almost the same". At this stage it is difficult to judge the solidity of these conclusions.

Response: We agree with the reviewer on the qualitative or semi-quantitative nature of the model predictions in this study. We carried out model simulation in an abstract 3D template as shown in Zhou et al., Science 2018. There is no bijection (one-to-one correspondence) between the cells in our model template and the cells of the real Arabidopsis SAM (imaged by confocal microscopy). Though establishing a quantitative model of the Arabidopsis SAM is of great interest, in our opinion, it is beyond the scope of our current work. Thanks for the comments. We have revised the main text and we described our simulation results as “qualitatively comparable to experimental observations” instead of “same” or “almost the same”.

Also some modeling data presented in the manuscript are difficult to understand. Fig 5 a-d: either I have missed something or what is done in the model is forcing the L1 input in all layers.

Response: We apologize that we did not provide clear description in the initial submission. In the revised manuscript, we have stated the values of model input (ATML1) in the SAM with 1.1 a.u. in the L1 and 0.4 a.u. in the L2 and corpus. We defined the values of the input in this way, because we were afraid that the 35S promoter driven gene was not ubiquitously/uniformly expressed in each layer of the SAM. Indeed, our in situ results shown in the Supplementary Fig. 10 in the revised manuscript are consistent with our consideration, and the pattern of ATML1 is also comparable to our setting of the input (Fig. 7 a, b).

And here the "Dex induction" (see my comment above: this needs to be changed) changes the values in L1 vs internal layers but has a mild effect on internal tissues. This is likely because the "induction" is mild and just additive in comparison to mock but this requires explanations. This is the case also for Fig S9: why are the 2 "mocks" and "Dex" different? The authors really need to significantly expand their description of the modeling work in the manuscript.

Thanks for the comments. We have revised the labels for figures of the modeling results. Based on the RNA in situ of ATML1 in 35S ATML1-GR, we found that ATML1-GR expression is not uniform and is qualitatively comparable to our current setting of the input (1.1 in the L1 and 0.4 in the L2 and corpus).

In the Fig. S9 in the initial submission, a & c are from the same simulation of miR171 in the wild type, and b & d are from the same simulation result of miR171 in the ATML1 ectopic activation line. The only difference between ab and cd is the value range of the color bar for the visualization (the range of 0-0.5 for a and b and the range of 0-6.84 for c and d). In the revised manuscript, we have included the new results in Supplementary Fig. 12, which better present our simulation results. In addition, we have significantly expanded the description of the modeling work in both the main text and the methods.

From the modeling work, one key expected outcome of the expression of ATML1 throughout the SAM is the induction of the MIR171 genes in all layers. This is nicely shown for MIR171B in Fig5 but for MIR171A Fig S8 raises quite a lot of question. The authors are surprisingly showing a zoom on one of the flower rather than showing the SAM and the induction seems far less efficient and patchy. The authors need to show what happens in the SAM. Rather than claiming

that the data perfectly support their model (Page 6 concluding phrase, last paragraphe), they should discuss the fact that the regulation might be more complex than what they claim (which is not a problem but a biological reality).

Response: We appreciate this comment. To carefully examine the induction of MIR171A, we performed a new experiment using the standard RNA in situ hybridization method (Supplementary Fig. 11), to avoid any side effects caused by potential photobleaching and photodamaging from the live imaging experiments. As shown in Supplementary Fig. 11, using the identical procedure, we performed the RNA in situ hybridization with the Mock and the Dex treated samples that were grown in the same condition and fixed at the same time. Our new results demonstrated that indeed MIR171A was greatly induced in the shoot apex one day after the Dex treatment in the 35S::ATML1-GR plant, which is consistent with our qPCR results (now shown in Fig. 7j). However, slightly different from the MIR171B, we found that in the SAM, the induction of MIR171A is not as efficient as that of 171B upon activation of 35S::ATML1-GR (with strong activation in young leaves but only mild and uneven induction in the SAM), especially at the deep cell layers of the SAM. In summary, we totally agree with the reviewer, and we think these results may reflect the biological reality or limitation of the current experimental system. Our results also suggest potential differences in the regulation of the MIR171A vs. MIR171B in distinct cell types in the SAMs, and the efficient induction of MIR171A in corpus may require other factors in addition to ATML1. We have included this discussion in the main text. Thanks for the understanding on the complexity of the biological system we work on.

Page 8 on de novo axillary stem cell niches, in the concluding paragraph the authors give the feeling that they have been able to confront the dynamical vision that they have in their simulation with the actual dynamics of the patterning of the lateral meristem. While this could be true for the establishment of the HAM gradient, the wet data do not allow to judge when mir171 is turned on in relation to ATML1 expression. The authors need to carefully check this entire section and its conclusion for accuracy.

Response: We thank the reviewer 2 for pointing out this concern, and for the understanding of the difficulty in precisely determining the temporal relation between miR171 and ATML1, and between mir171 and HAM. We totally agree with the reviewer. Because our experimental results do not allow us to evaluate any simulated timing on when and how long ATML1 turns on miR171 nor miR171 turns off HAM during the continuous development of the AM, simulation of the dynamics of new meristem initiation may not be very informative or even misleading.

In the revised manuscript, we chose not to perform the simulation of this continuous dynamic process in AMs, and we no longer claimed that we can perfectly simulate the dynamics of different gene expression during the continuous development of the axillary meristem. Instead, we only reported what we have seen from the RNA in situ hybridization experiments in different samples at different developmental stages, suggesting that the L1-miR171-HAM signaling cascade is also active in the de novo stem cell niches. We again appreciate the reviewer's comments that helped us better present our work.

Other comments:

- P2 top of the page: "HARIY MERISTEM" should be "HAIRY MERISTEM"

Response: Thanks and we have revised it in the main text.

- Page 6 middle of the page: "ATML1 directly activates 171 in the SAM"; 171 should probably be miR171.

Response: Thanks and we have revised it in the main text.

- End of Page 6: "... Dex treatment compared with mock control (Fig. 5)." Please specify the panel(s) on Fig 5 that is(are) concerned.

Response: Thanks and we have revised it in the main text.

- Page 8 1st paragraph of Discussion: "MIR171A and MIR171B genes are the major contributor for miR171 activity" - this has not been demonstrated by the authors; either they need to provide data or this needs to be corrected.

Response: Thanks for the comment. Through the confocal imaging with the identical setting, we found that *MIR171A* and *MIR171B* reporters are highly and specifically expressed in the epidermis of the SAM, leaf primordia and floral primordia (Fig. 1 a-b). In contrast, the *MIR171C* and *MIR170* reporters are expressed at a much lower level in the epidermis of SAMs, and the signal from these two reporters is not detectable in the young leaves (Fig. 1 c-d). Based on these observations, we included a discussion that *MIR171A* and *MIR171B* genes are likely the major contributors for miR171 activity in SAMs and SAM derived leaves and flower meristems.

- Page 10 last paragraph of discussion: "ATML1-miR171-HAM" - elsewhere this module is called L1-miR171-HAM which is probably more accurate as the authors suggest a role for PDF2. This needs to be corrected and the authors need to ensure that they use the same name throughout the manuscript.

Response: Thanks and we have revised it in the main text.

Reviewer #3 (Remarks to the Author):

The authors study the role of microRNA within the apical-basal patterning in the SAM. This is an interesting problem well beyond the specific plant field. In this review I focus on the computational part of the manuscript and I leave the experimental part to others. I have a few major problems with this part:

1. The authors put the model quite prominently in the main text ("new 3D computational

model"). However, they fail to sufficiently describe what they have done. Either they extend and clarify this part or it needs to be removed. It cannot be to the reader of a sound scientific paper to guess what was done. I suggest a cartoon like figure together with some, e.g., pseudo code in the MAIN text. The simulation details also need to be described somewhere and not in a previous paper.

Response: We appreciate the reviewer for pointing out this concern. In the revised manuscript, we have described this model in details both in the main text and in the methods.

2. The equations in the Methods section are insufficiently explained.

Response: Thanks for the comment. In the revised manuscript, we have significantly expanded our model descriptions, which explain every equation and all the parameters in detail. We also described all the assumptions or conditions we have made when building up this model and the rationales and experimental evidence for these assumptions or conditions.

Why is the activation of [HAMr] modelled in this way?

Response: We thank the reviewer for asking this question, which promoted us to carefully think about how to model the activation of [HAMr] in a way that both biologists and computer scientists can understand well. In our revised version, we revised and simplified this term, and we defined a constant activation process in all the cells from different cell layers. Consistently, our pHAM2::H2B-GFP reporter marker shows high expression in every single cell from different cell layers of the SAM, including epidermis, sub-epidermis, and corpus. We also want to point out that the simulation results using the equations listed in the previous submission and the equations in the revised manuscript here are largely comparable, and they lead to the same conclusion.

What does the Delta mean in equation 1? The Laplacian operator? How is it discretised?

Response: D_{mir171} is the diffusion constant of miR171 RNA among SAM cells. Δ is the Laplace operator, and the contribution of $\Delta[miR171]$ to the derivative for miR171 RNA in cell i is given by $\sum_n([miR171]_n - [miR171]_i)$, where the sum is over neighboring cells. The set of neighbors is more restricted at boundaries. We have included these descriptions in the methods. Again, we thank the reviewer for helping us explain our model more clearly.

What about boundary conditions?

Response: We focus on the gene expression pattern in the SAM, but to date, there is no clear experimental evidence for fluxes of the gene products (*ATML1*, *miR171*, and *HAM*) into and/or out of the Arabidopsis SAM. Therefore, the boundary condition for our current model is set to be no flux boundary, similar to the boundary conditions defined in our previous model (Zhou et al., 2018).

3. I severely doubt the delay in equation 2 (degradation). I understand the intention, but I doubt that the implementation is correct. At least it is not consistent with the description the authors give. It is NOT that it takes a while after complex formation until [HAMr] is removed from the system. It rather models a maturation time of [mir171], which may not what the authors want to model. A delayed removal would rather have a delay in both, [HAMr] and [mir171]. If the authors believe I am wrong they should provide for me and the interested reader a derivation of the delay in equation 2.

Response: We thank the reviewer for the critical comment and for the understanding of the difficulty in precisely determining the temporal relation between mir171 and ATML1, and between mir171 and HAM. We totally agree with the reviewer that there is no direct evidence to prove or disprove a delay for the repression of HAM by miRNAs, although we think a delay is possible. In the revised manuscript, we remove the term of the delay in the equation, and the new simulation results in the SAM (Fig. 7) are very similar to the ones presented in the initial submission.

In addition, we agree with the reviewer 2 that the simulation of dynamics of new meristem initiation is not very informative or even misleading. We chose not to perform any simulation of this continuous process in AMs, because our experimental results do not allow us to evaluate any simulated timing on when and how long ATML1 turns on miR171 nor miR171 turns off HAM during the continuous growth and development. Taken together, we thank both reviewers for the comments that helped us better present our work.

4. Where do the parameters come from? How certain are the authors about them and why? What about a sensitivity analysis? Usually these parameters are only defined in an uncertainty range. How does the model behave within the hypercube defined by all parameter ranges? A local analysis, based on parameter fiddling is not good enough here.

Response: Thanks for this critical comment. We totally agree that a local analysis with a set of fixed parameters is not enough, and thus, we worked out a number of solutions with different parameter values to further test the model.

Specifically, our model contains six key parameters that control the dynamics of miR171 and the HAM mRNA expression pattern (Supplementary Tables 1 and 2). We have explored the working ranges of these six key parameters (Supplementary Table 2). In total, we defined 235 sets of different combinations of parameter values (Supplementary Table 3). Using each set of parameter values, we can simulate patterns of HAM mRNA that are qualitatively comparable to the observed pattern in the WT SAM (shown as Supplementary Movies 7-12). Please note that all the simulation work with different parameter values has been performed in the 3D template, and importantly, these tests demonstrated that the system we established is robust.

Furthermore, using these 235 sets of parameter values, we carried out sensitivity analysis for these six key parameters (Supplementary Fig. 15) following the published procedure (Yadav et al., *Molecular Systems Biology* 2013). In the analysis, the parameters were increased by 1%. The sensitivity was computed as: $\frac{\Delta m}{\Delta p} \times \frac{p}{m}$. p is the original parameter value.

Δp is the variation of the parameter. m is the total *HAM* mRNA expression. Δm is the variation of the total *HAM* mRNA expression. In general, the model output shows robust behavior to perturbations in the six parameters. *HAM* mRNA expression level is more sensitive to k_{hrp} and k_{hrnh} , which directly control its production and degradation.

5. The relevant literature on modelling approaches of the SAM are not sufficiently mentioned nor cited.

Response: Thanks for the comment. In the revised manuscript, we have included the citations of the previously reported modeling work that is related to gene expression patterns in the SAM in the methods.

As I said initially, I focus on the modelling part, as I have here severe doubts that this work lives up to the standard of Nature Communications. Nevertheless, the experimental part may do and a way out for the authors may be to remove the modelling part. Or to address my comments above.

Response: We sincerely appreciate the reviewer for a number of constructive suggestions and comments on the modeling work. To address all the reviewer's comments above, we have included detailed descriptions of the model structure, equations, assumptions, and parameters in main text and in the methods. We have revised two terms in the equations for more precisely computing the agent dynamics without affecting the results of simulation. We have examined and defined the working ranges of six key parameters that control dynamics of miR171 and the *HAM* mRNA expression pattern, and we worked out 235 solutions with different parameter values. We have performed the sensitivity analyses for these six parameters using all 235 sets of solutions. We have also cited more literature of the previously related modeling work. In our opinion, the modeling work in the manuscript has been significantly improved and better described. If the reviewer has any additional concern and/or suggestion, we will be very happy to consider and continue to improve our modeling work.

Reviewers' comments:

Reviewer #1 (Remarks to the Author):

The authors added data concerning the binding of AtML1 and PDF to the miR171 promoters and the expression of the miR170/171 in roots in order to address the previous criticism. They also rely less heavily on the computational model and add more detailed descriptions for the model. Apart from that, the core statement of the paper still is the regulation of HAM by miR171 which is induced by ATML1/PDF2 in the L1 of the shoot meristem. I have doubts about the novelty of this concept. The authors claim that their approach to the expression pattern of miR171 is broader and deeper in scope than the one published in Takanashi et al., 2018. Nevertheless, they only show that the homologous miRNA genes have an expression similar to the reported miR171A pattern. Furthermore, as HAM was described before to continue functioning from the embryonic SAM through all developmental stages, the revelation, that miR171 also shows the same expression pattern in seedling SAMs and in axial meristems, seems obvious. In connection to that, the involvement of ATML1/PDF2, though not shown before, is hardly surprising, as several publications have already shown ATML1/PDF2 to be responsible for the L1-exclusive expression of a number of factors, even of itself (Abe et al., 2001; Abe et al., 2003; Nakamura et al., 2006; Takada and Jürgens, 2007).

The authors added a picture of an in situ hybridization experiment purporting to demonstrate the movement of miR171 into the deeper meristem layers. Unfortunately, it is hard to judge without a matching negative control, that the authors neglect to provide. Even taken at face value, it seems that there's strong miR171 signal in the L1 and evenly weak signal in the whole remaining meristem. I disagree with the authors that this is in agreement with their HAM2 transcriptional and translational reporters, as the expression of the latter begins in lower corpus layers than you would expect from the in situ hybridization. The in situ also is in contrast with the input they used in their computational model, where they "allowed miR171 to move into the L2 and upper corpus layers in a diffusion-like manner." Again it's hard to judge without negative control, but miR171 should move into the whole corpus area.

In summary, although the authors made small changes, I do not think that the paper provides sufficient novelty.

Reviewer #2 (Remarks to the Author):

The authors have globally addressed my concerns satisfactorily and the revised version is in my view rather solid. The authors have notably largely expanded their description of the modeling work. It is now presented in a way that makes the combined use of modeling and wet experiments easier to understand and much more accurate. They have also explored the parameter space in their model and show that there are little changes in the virtual pattern of HAM.

However there is still a few points on which I think the authors should be a bit more careful and/or provide a better justification:

- Can the authors justify the range used for the parameter exploration in the model? They should also better explain in the text what they conclude from the sensitivity analysis. I doubt that this is obvious to most potential readers.

- "Response: We apologize that we did not provide clear description in the initial submission. In the revised manuscript, we have stated the values of model input (ATML1) in the SAM with 1.1 a.u. in the L1 and 0.4 a.u. in the L2 and corpus." Why is it 1.1/0.4 vs 1/0 in wt? This is not a crucial point but the choice of parameter seems difficult to understand here.

- "Response: Thanks for the comment. Through the confocal imaging with the identical setting, we found that MIR171A and MIR171B reporters are highly and specifically expressed in the epidermis of the SAM, leaf primordia and floral primordia (Fig. 1 a-b). In contrast, the MIR171C and MIR170 reporters are expressed at a much lower level in the epidermis of SAMs, and the signal from these two reporters is not detectable in the young leaves (Fig. 1 c-d). Based on these observations, we

included a discussion that MIR171A and MIR171B genes are likely the major contributors for miR171 activity in SAMs and SAM derived leaves and flower meristems." Expression levels might not be the best indicator of the activity of a microRNA; e.g. although expressed at lower levels a microRNA might have a higher affinity. Also the authors show that MIR171C and MIR170 are target of ATML1 using Y1H. Thus the authors do not have really strong argument to say that MIR171C and MIR170 are less important than MIR171A,B. The need to revise the text accordingly. Notably, a phrase like the last one lines 307-308 cannot be justified even by their own data.

A few typos to correct:

Line 30: "mircoRNAs" should be "microRNAs"

Line 71: "the additional setting" should be "additional settings"

Reviewer #3 (Remarks to the Author):

I find the manuscript significantly improved. As I focused in my previous review on the modelling part, I will only refer to this here. The authors invested some work to enhance and improve the modelling sections in the main text and in the supplement. I am almost satisfied, but I have a few points:

1. the description of the model in the main text is good but should be complemented by a schematic. This will make it much easier for the reader to understand the model.
2. the model equations should be presented together without text in between. The learned reader knows how to read equations as a concise description of the model. The text can follow after the equations and needs only to clarify unclear points (such as the implicit spatial dependence of some parameters, etc.)
3. it should be mentioned what [ML1p] is
4. it is ok to set [ML1p]=1 in the L1 layer and 0 otherwise, but I do the treatment of the perturbation of the ectopic activation not very convincing. Why 1.1 and 0.4 and not 1.3 and 0.5? Probably the simulation results do not depend qualitatively on the exact values, but SOMETHING must be important here. The authors should clarify this, also to remove the taste of complete arbitrariness.
5. in my opinion the sampling of the parameters is insufficient: using an intervall of 40% - 200% around an arbitrarily chosen parameter is not good enough. It is valid to focus on the 6 relevant parameters, but if one is ignorant about the true parameter values, it is good practice to set a lower bound (usually zero) and an upper bound (based on general reasoning about, e.g., protein binding, stability, etc.) and then sample uniformly on the log-scale, preferably done using quasi Monte Carlo or (second best choice) Latin Hypercube. The conclusions drawn from the model may not change, but it is simply about good scientific practice as one would also expect it for experiments.

In conclusion, I find the manuscript improved, but I am still not convinced that the modelling part has the standard needed for a journal as Nature Communication.

Reviewers' comments:

Reviewer #1 (Remarks to the Author):

The authors added data concerning the binding of AtML1 and PDF to the miR171 promoters and the expression of the miR170/171 in roots in order to address the previous criticism. They also rely less heavily on the computational model and add more detailed descriptions for the model.

Response: Thanks for the comments. We are happy to know that we have largely improved our manuscript.

Apart from that, the core statement of the paper still is the regulation of HAM by miR171 which is induced by ATML1/PDF2 in the L1 of the shoot meristem. I have doubts about the novelty of this concept. The authors claim that their approach to the expression pattern of miR171 is broader and deeper in scope than the one published in Takanashi et al., 2018. Nevertheless, they only show that the homologous miRNA genes have an expression similar to the reported miR171A pattern. Furthermore, as HAM was described before to continue functioning from the embryonic SAM through all developmental stages, the revelation, that miR171 also shows the same expression pattern in seedling SAMs and in axial meristems, seems obvious. In connection to that, the involvement of ATML1/PDF2, though not shown before, is hardly surprising, as several publications have already shown ATML1/PDF2 to be responsible for the L1-exclusive expression of a number of factors, even of itself (Abe et al., 2001; Abe et al., 2003; Nakamura et al., 2006; Takada and Jürgens, 2007).

Response: Thanks for the comments.

First, it is significant to define that the expression of *MIR171B*, *MIR171C* and *MIR170* genes is all confined in the epidermis of the SAMs. Different homologous genes do not necessarily have identical expression domains, and indeed the expression patterns of distinct family members can be completely different. One such example is the well characterized WUS/WOX family. If one or more MIR171/170 genes were specifically expressed in other cell layers (corpus, for example) of the SAM, undoubtedly our current working hypothesis would have to be changed. Thus, our finding that all MIR171 genes are specifically expressed in the epidermis of SAM is both novel and important. Further, the expression domain of the MIR171 genes only represents a very small portion of our findings presented in the manuscript, composing one figure (Figure 1) out of a total of eight main figures and 18 supplementary figures.

Second, it is significant to define that ATML1 directly binds the promoters of MIR171 genes and directly activates MIR171 in the SAM. The reviewer cited a few factors whose L1-exclusive expression is determined by ATML1/PDF2 (Abe et al., 2001; Abe et al., 2003; Nakamura et al., 2006; Takada and Jürgens, 2007). We have known these findings and cited them in our manuscript. However, reports have shown that hundreds of Arabidopsis genes are differentially or specifically expressed in L1, based on the fluorescence assist cell sorting and /or TRAP seq

experiments (Tian et al., Nature Communications 2019, Yadav et al., Development 2014). Can ATML1 directly bind to the promoters of all these genes? Can ATML1 directly activate the expression of all of them in the SAM? Apparently, knowing a few L1-specific factors that are directly regulated by ATML1 does not imply the regulatory mechanism of MIR171. Without our biochemical, live imaging and molecular genetic experiments, it is not possible to conclude the direct relationship between ATML1 and MIR171, nor uncover the molecular mechanism that initiates and maintains the apical-basal pattern of HAM. These findings compose a major part of our manuscript.

The authors added a picture of an in situ hybridization experiment purporting to demonstrate the movement of miR171 into the deeper meristem layers. Unfortunately, it is hard to judge without a matching negative control, that the authors neglect to provide. Even taken at face value, it seems that there's strong miR171 signal in the L1 and evenly weak signal in the whole remaining meristem. I disagree with the authors that this is in agreement with their HAM2 transcriptional and translational reporters, as the expression of the latter begins in lower corpus layers than you would expect from the in situ hybridization. The in situ also is in contrast with the input they used in their computational model, where they "allowed miR171 to move into the L2 and upper corpus layers in a diffusion-like manner." Again it's hard to judge without negative control, but miR171 should move into the whole corpus area.

Response: Thanks for the comments.

First, we would like to point out that although we included one picture of the miR171 in situ result, we have repeated and confirmed this result with five independent biological replicates (please see the figure legend and methods for this information).

Second, we agree that a negative control can help confirm the pattern of endogenous miR171. Because a murine miR124 LNA probe has been suggested and demonstrated as a suitable negative control for microRNA in situ experiments in plants (Javelle and Timmermans, Nature protocols 2012, Douglas et al., Plant Cell 2010), we used this probe (Qiagen) as our negative control. We performed new RNA in situ experiments using the miR171 LNA probe side by side with the murine miR124 LNA probe using the identical procedure. The wild type plants grown in the same condition and harvested at the same time were used for the miR171 in situ and the murine miR124 in situ experiments (Supplementary Figure 8). The results shown in Supplementary Figure 8 were repeated and confirmed with four biological replicates for each probe.

Compared to the negative control, our new results (Supplementary Figure 8) are consistent with our previous ones (Supplementary Figure 9), showing that miR171 is present not only in the L1 layer where it is synthesized (Fig. 1), but also a couple of layers below the L1 at a lower level. Note that RNA in situ hybridization, miRNA in situ particularly, is not quantitative enough to define an apical-basal concentration gradient of miR171 in the SAM, especially when the level of miR171 is very low at deep layers. Our miR171 in situ results qualitatively agree with the expression pattern of the miR171 sensitive HAM2 translational fluorescence reporter

(Supplementary Movie 5), which is completely off in epidermis, greatly repressed in sub-epidermis, and highly expressed in corpus. All these results together point out that in the SAM, miR171 is specifically expressed in epidermis, and it moves down through limited number of cell layers and shapes the HAM gradient.

Reports have shown that in the SAM, microRNAs only move through limited distance. For example, when expressed in tunica (L1 and L2) layers of the SAM, the movement of an artificial miRNA is limited, moving down only one cell layer (Skopelitis et al., Nature Communications 2018). As another example, miR394b that is transcribed in the L1 layer of the SAM forms a concentration gradient with the movement through one or two cell layers (Knauer et al., Developmental Cell 2013). As we have discussed in the main text, our results together with all the previous reports support an idea that the movement of microRNAs through limited number of cell layers in the SAMs is likely an important feature for microRNAs to function.

Please also note that our in situ results (Supplementary Figs. 8, 9) are indeed consistent with our computational model input and simulation. In the model, we set a very low diffusion constant for the movement of miR171 (Supplementary Table 1). Please also take a look at the simulated miR171 pattern in the SAM from the Supplementary Figure 16 a (that is Supplementary Figure 12 a in the previous version of the manuscript), which is qualitatively comparable to the pattern from our in situ results (Supplementary Figure 8 and Supplementary Figure 9).

Taken together, both computational simulation and experimental results show that the limited movement of miR171 in the SAM from epidermis to deep layers is sufficient to generate the apical-basal concentration gradient of HAM (Fig. 7m, Supplementary Movie 5, 9, 13). We also revised the description in the main text to “we set a low diffusion constant for the movement of miR171 from epidermis to deep layers” to better describe this process.

In summary, although the authors made small changes, I do not think that the paper provides sufficient novelty.

Response: Thanks for the comment. We think our manuscript provides sufficient novelty and is of broad interest.

Along axis of multicellular organisms, the apical-basal polarity of either gene expression or protein localization is crucial for cell patterning and the developmental decision making. How this apical-basal polarity is initiated and maintained through the continuous cell division and growth is an important and long-lasting question. Our work presented here tackled this question from one perspective, using Arabidopsis SAMs as a model. Our mechanistic study provides a new framework showing that epidermis exclusive transcription factors determine the apical-basal polarity of key meristem regulators in multiple cell layers, a machinery applicable to various shoot meristems (SAM, FM and AM). Specifically, we experimentally defined and computationally simulated a L1-miR171-HAM signaling cascade that travels along axis of the SAM and regulates meristem function.

Reviewer #2 (Remarks to the Author):

The authors have globally addressed my concerns satisfactorily and the revised version is in my view rather solid. The authors have notably largely expanded their description of the modeling work. It is now presented in a way that makes the combined use of modeling and wet experiments easier to understand and much more accurate. They have also explored the parameter space in their model and show that there are little changes in the virtual pattern of HAM.

Response: Thanks for the comment! We are happy to know that we have largely improved our manuscript.

However there is still a few points on which I think the authors should be a bit more careful and/or provide a better justification:

- Can the authors justify the range used for the parameter exploration in the model?

Response: Thanks for this question. In the last version of our manuscript, we have explored a small range of numbers for the six key parameters (Supplementary Table 2), focusing on values in a range from 40% to 200% of each fixed parameter value (shown in Supplementary Table 1 and Supplementary Table 2). In total, we defined 235 sets of different combinations of parameter values within this range (Supplementary Table 3). Using each set of parameter values, we can simulate the expression patterns of *HAM* mRNA that are qualitatively comparable to the observed pattern in the wild type SAM (shown as Supplementary Movies 7-12). In the current version of revised manuscript, we further analyzed and explored these six parameters. Following the suggestions from Reviewer 3, we have performed unbiased random search using quasi Monte Carlo method to look for more sets of parameters that can lead to qualitatively comparable *HAM* mRNA patterns in the wild type SAM. In total, we have found 173 new sets of different parameters (Supplementary Table 5) out of 20,000 sets of random parameters (Supplementary Table 4). The simulation using each of these 173 sets of parameters is able to qualitatively represent the signaling cascade (*HAM* mRNA pattern) that we tested experimentally (Supplementary movie 13), suggesting that our system is robust. We have included the new results and detailed descriptions in the manuscript.

They should also better explain in the text what they conclude from the sensitivity analysis. I doubt that this obvious to most potential readers.

Response: Thanks for the suggestion. We have revised the manuscript, and we have explained results from both the previous sensitivity analysis (Supplementary Figure 10) and the new sensitivity analysis (Supplementary Figure 11) in the main text. In summary, we found that among six parameters, the total *HAM* mRNA amount is more sensitive to the changes in three of them, including k_{hrp} and k_{hrnh} that directly control the production and degradation of *HAM* mRNA, respectively, and k_{mirn} that directly determines the degradation of miR171. We believe these additions will help readers better understand our analyses and conclusion.

- "Response: We apologize that we did not provide clear description in the initial submission. In the revised manuscript, we have stated the values of model input (ATML1) in the SAM with 1.1 a.u. in the L1 and 0.4 a.u. in the L2 and corpus." Why is it 1.1/0.4 vs 1/0 in wt? This is not a crucial point but the choice of parameter seems difficult to understand here.

Response: Thanks for bringing up this question. The input values of 1.1 a.u. / 0.4 a.u. in the ATML1 ectopic activation line vs 1 a.u. / 0 a.u. in wild type are defined based on the results from RNA *in situ* experiments shown in the Supplementary Figure 12 (Supplementary Figure 10 in the last manuscript). We first looked at *ATML1* in the L1 layers, and the level in the *35S::ATML1-GR* line looks slightly higher than that in the wild type. Therefore, we assigned 1.1a.u. for ML1p input in L1 for the *35S::ATML1-GR* line. We then looked at the deeper layers, and the level of *ATML1* in the *35S::ATML1-GR* line is lower than that in the L1 layer of the same plant but higher than that in the deeper layers of the wild type plant. Therefore, we assigned 0.4 a.u. for the ML1p input in deep layers for the *35S::ATML1-GR* line.

We understand that these values are not based on quantitative analyses. The RNA *in situ* experiments we performed can only provide qualitative or at most semi-quantitative information. The 1.1 a.u. / 0.4 a.u. in the ATML1 overexpression line vs 1 a.u. / 0 a.u. in wild type reflect the experimental observation in the model and are qualitatively comparable to the patterns shown from the *in situ*.

Furthermore, we explored a broader possible range of the ML1p input values (L1: 1.05-1.5 a.u., deep layers: 0.3-0.6 a.u.). 25 new input values all lead to the suppression of *HAM* mRNA expression in deep layers of SAMs (Supplemental Movie 14), largely comparable to the simulation result using our initial input values (1.1 a.u. / 0.4 a.u.). All these results support our conclusion that this model can qualitatively reflect the signaling cascade we proposed in the SAM. We have included these discussions and new results in the current manuscript.

- "Response: Thanks for the comment. Through the confocal imaging with the identical setting, we found that MIR171A and MIR171B reporters are highly and specifically expressed in the epidermis of the SAM, leaf primordia and floral primordia (Fig. 1 a-b). In contrast, the MIR171C and MIR170 reporters are expressed at a much lower level in the epidermis of SAMs, and the signal from these two reporters is not detectable in the young leaves (Fig. 1 c-d). Based on these observations, we included a discussion that MIR171A and MIR171B genes are likely the major contributors for miR171 activity in SAMs and SAM derived leaves and flower meristems." Expression levels might not be the best indicator of the activity of a microRNA; e.g. although expressed at lower levels a microRNA might have a higher affinity. Also the authors show that MIR171C and MIR170 are target of ATML1 using Y1H. Thus the authors do not have really strong argument to say that MIR171C and MIR170 are less important than MIR171A,B. The need to revise the text accordingly. Notably, a phrase like the last one lines 307-308 cannot be justified even by their own data.

Response: Thanks for pointing out this concern. We agree with the reviewer and we have revised our discussion in the main text. Our results suggest that these four genes are responsible for the miR171 activity in the SAM and SAM derived leaves and flower meristems, and *MIR171A* and *MIR171B* genes contribute more to the total miR171 level in the SAM.

A few typos to correct:

Line 30: "mircoRNAs" should be "microRNAs"

Line 71: "the additional setting" should be "additional settings"

Response: Thanks! We have revised them in the main text.

Reviewer #3 (Remarks to the Author):

I find the manuscript significantly improved. As I focused in my previous review on the modelling part, I will only refer to this here. The authors invested some work to enhance and improve the modelling sections in the main text and in the supplement. I am almost satisfied, but I have a few points:

1. the description of the model in the main text is good but should be complemented by a schematic. This will make it much easier for the reader to understand the model.

Response: Thanks for the suggestion. We have included a schematic diagram describing the model (Supplementary figure 7) in the current version of the manuscript.

2. the model equations should be presented together without text in between. The learned reader knows how to read equations as a concise description of the model. The text can follow after the equations and needs only to clarify unclear points (such as the implicit spatial dependence of some parameters, etc.)

Response: Thanks for the suggestion. We have made the changes accordingly.

3. it should be mentioned what [ML1p] is

Response: Thanks for the suggestion. In our model, [ML1p] refers to the concentration of functional ATML1 protein and its homolog PDF2 protein. This information is now included in the main text and the methods of the revised manuscript.

4. it is ok to set [ML1p]=1 in the L1 layer and 0 otherwise, but I do the treatment of the perturbation of the ectopic activation not very convincing. Why 1.1 and 0.4 and not 1.3 and 0.5? Probably the simulation results do not depend qualitatively on the exact values, but SOMETHING must be important here. The authors should clarify this, also to remove the taste of complete arbitrariness.

Response: Thanks for the critical comments. The input values of 1.1 a.u. / 0.4 a.u. in the ATML1 ectopic activation line vs 1 a.u. / 0 a.u. in wild type are defined based on results from RNA *in situ* images shown in the Supplementary Figure 12 (Supplementary Figure 10 in the last manuscript). We first looked at ATML1 in the L1 layers, and the level in the 35S::ATML1-GR line looks slightly higher than that in the wild type. Therefore, we assigned 1.1 a.u. for ML1p input in L1 for the 35S::ATML1-GR line. We then looked at the deeper layers, and the level of ATML1 in the 35S::ATML1-GR line is lower than that in the L1 layer of the same plant but higher than that in the deeper layers of the wild type plant. Therefore, we assigned 0.4 a.u. for ML1p input in deeper layers for the 35S::ATML1-GR line. We understand that these values are not based on quantitative analyses. Considering the fact that the current RNA *in situ* technique generally provides us qualitative or at most semi-quantitative information, it is hard to precisely quantify these levels. The 1.1 a.u. / 0.4 a.u. in the ATML1 overexpression line vs 1 a.u. / 0 a.u. in the wild type reflect the experimental observation in the model and are qualitatively comparable to the patterns shown from the *in situ*.

We also think other ML1p input values need to be further explored. Based on our *in situ* images, we defined the possible range of ML1p patterns to 1.05-1.5 a.u. for the L1 layer and 0.3-0.6 a.u. for the deep layers. Within this range, we randomly explored 24 different ML1p input patterns, and carried out simulations using these randomly defined ML1p inputs. The input value of 1.3 a.u. / 0.5 a.u. is also included for the simulation. We found all these 25 ML1p inputs lead to the suppression of the HAM mRNA expression in deep layers of SAMs, which are largely comparable to the simulation result using the initial input value (1.1 a.u. / 0.4 a.u.) (Supplemental Movie 14). All these results support our conclusion that the computational model can qualitatively reflect the signaling cascade that we proposed in the SAM. We have included the new results and discussions in the current version of revised manuscript.

5. in my opinion the sampling of the parameters is insufficient: using an interval of 40% - 200% around an arbitrarily chosen parameter is not good enough. It is valid to focus on the 6 relevant parameters, but if one is ignorant about the true parameter values, it is good practice to set a lower bound (usually zero) and an upper bound (based on general reasoning about, e.g., protein binding, stability, etc.) and then sample uniformly on the log-scale, preferably done using quasi Monte Carlo or (second best choice) Latin Hypercube. The conclusions drawn from the model may not change, but it is simply about good scientific practice as one would also expect it for experiments.

Response: We really appreciate these critical and constructive comments! We totally agree that it is very good practice to explore a larger range of independent parameters through random searching. Following the suggestions, we have performed the unbiased random search using quasi Monte Carlo method to look for more sets of parameters that can be used in our model. In total, we have found 173 new sets of different parameters out of 20,000 sets of random parameters. The simulation using each of these 173 sets of parameters is able to qualitatively represent the signaling cascade (HAM mRNA pattern) that we tested experimentally (Supplementary movie 13).

We first defined the lower bound and the upper bound of each parameter for the random search, based on knowledge of biological context in general and specific features of the regulatory system in the SAM being studied. Because a wild type SAM is dynamically stable, the production and degradation of the same agent (RNA/protein) need to be balanced. Therefore, we set the production and degradation rates for the agent to vary within a 100 fold range. In addition, because the miR171 regulates the HAM mRNA levels through direct binding and cleavage, we set the difference of turnover rates for these two agents within a 100 fold range. With these considerations, we performed the search for parameter values in the range of 0.1-1 (arbitrary unit) for miR171 production rate, miR171 degradation rate, HAM mRNA production rate, miR171 dependent HAM degradation rate, and miR171 independent HAM mRNA degradation rate. Further, because movement of microRNAs between cells in the SAM is generally limited (Skopelitis et al., Nature communications 2019), we defined the search for parameter values of miR171 diffusion constant in the range of 0.001-0.1 (arbitrary unit).

Within the ranges we defined, we used Quasi Monte Carlo Method to randomly sample 20,000 sets of six different parameters uniformly on the log-scale. We then ran simulations using these 20,000 random parameter sets (shown in Supplementary Table 4) in the 3D template that we have been using to find more solutions for HAM mRNA pattern in the SAM. After carefully comparing these simulation results, we found 173 new sets of solutions (shown in Supplementary Table 5), which are different from the initially selected set of parameters (shown in Supplementary Table 1). Furthermore, we carried out new sensitivity analysis using these 173 sets of new parameters (Supplementary Figure 11). Compared with previous results from the sensitivity analysis based on 235 “local” sets of parameters (Supplementary figure 10), the new sensitivity analysis results based on the 173 random parameter sets show the same tendencies for the parameter sensitivities.

In conclusion, I find the manuscript improved, but I am still not convinced that the modelling part has the standard needed for a journal as Nature Communication.

Response: Thanks for the comments. In the current version of revised manuscript, we have included the schematic figure to help describe the model (Supplementary Figure 7), and we revised the presentation of equations and descriptions in the methods for better clarity. We provide better explanations on how we defined the ML1p input in the ATML1 overexpression line for the simulation, and we explore the effect of the input with new simulation results (Supplementary Movie 14). Further, we have largely expanded the range of the solutions and parameters. We have found 173 new sets of parameters (Supplementary Table 4) from 20,000 sets of randomly sampled parameters (Supplementary Table 5), which are different from the initial set of parameters used for the simulation. Using each of these 173 new sets of parameters, the model is able to simulate *HAM* mRNA pattern that is qualitatively comparable to the experimental observation (Supplementary Movie 13). We have performed additional sensitivity analysis using all sets of parameters that we have defined from different methods (Supplementary Figure 11), showing the same tendencies of parameter sensitivity (Supplementary Figure 10). Taken together, we believe our manuscript is significantly

improved, and we sincerely appreciate all the comments and suggestions from Reviewer 3 that helped us to make these improvements.

REVIEWERS' COMMENTS:

Reviewer #1 (Remarks to the Author):

In this newest version, the authors address some of my earlier concerns with the manuscript, especially by providing new in situ data in Supplemental Figure 8. As far as I can judge, the data is sound and evidence for the conclusions is sufficiently provided.

In the rebuttal to my last criticisms, the authors argued that it can neither be taken for granted that the MIR171/170 genes share the same expression domain nor that ATML1 is responsible for the expression of all L1-specific genes. While this is certainly true, it doesn't change the fact that the hypothesis posited by their earlier publications and bei Takanashi et al., 2018 is not changed by putting ATML1 on top of the already published MIR171 regulation in the present manuscript and that there is no other plant factor (apart from ATML1 co-factor PDF2) known at the moment that induces L1-specific expression in plants.

Because of these concerns I still have doubts about the claim of novelty, as in my opinion the manuscript confirms the framework already established in the aforementioned publications and expands on it by adding the only established L1 factor on top. As such I regret that I have to say that in my opinion the manuscript does not fit into the scope of Nature communications.

Because of these concerns I still have doubts about the claim of novelty, as in my opinion the manuscript confirms the framework already established in the aforementioned publications and expands on it by adding the only established L1 factor on top. As such I don't believe the manuscript fits into the scope of Nature communications.

Reviewer #2 (Remarks to the Author):

The authors have addressed satisfactorily all my remaining concerns.

Reviewer #3 (Remarks to the Author):

The authors responded to all my points in a satisfying manner. I find the manuscript further improved and have now only very minor comments:

1. The chosen values for ML_1p : I appreciate the work done by the authors to clarify this point. However, I find it still not 100 % satisfying. Maybe the authors could consider whether the difference between L1 and deeper layers is important for the functioning of the system. E.g. 1.1 and 0.4 works, and also 1.3 and 0.5. But would 1.1 and 0.7 also work? It may be the relative difference between these two values which matter. If this is true, I think the authors should state this in the manuscript.
2. The authors should not call $D_{\{mir171\}}$ the diffusion constant. A diffusion constant has units $Length^2/Time$ and $D_{\{mir171\}}$ has the unit $1/Time$. Maybe one could call it rescaled diffusion constant (rescaled by the intrinsic length, probably cell size) or transport rate.
3. The symbol Δ should not be called Laplace operator, as this is used for the differential operator. Rather, one could call it discrete Laplace operator, or alike.
4. When the authors explain the quasi Monte Carlo sampling and the chosen intervals the talk about arbitrary units, which I do not understand. I think all parameters have a unit, unless the authors non-dimensionalized them. In fact, most of the parameters used by the authors in the equations 1-5 have probably the dimension $1/h$.

Beside these minor points I have no further objection against publication of this manuscript.

Reviewer #3 (Remarks to the Author):

The authors responded to all my points in a satisfying manner. I find the manuscript further improved and have now only very minor comments:

Response: We really appreciate all the constructive comments and detailed suggestions from Reviewer 3. They helped us greatly improve the manuscript.

1. The chosen values for $ML1p$: I appreciate the work done by the authors to clarify this point. However, I find it still not 100 % satisfying. Maybe the authors could consider whether the difference between L1 and deeper layers is important for the functioning of the system. E.g. 1.1 and 0.4 works, and also 1.3 and 0.5. But would 1.1 and 0.7 also work? It may be the relative difference between these two values which matter. If this is true, I think the authors should state this in the manuscript.

Response: Thanks for this question.

In our last submission, we defined the possible expression patterns of *ATML1* based on our RNA in situ experiment (L1: 1.05-1.5; Deep Layers: 0.3-0.6). We then explored 25 different sets of $[ML1p]$ within this range and found that all the simulation results are comparable (Supplementary Movie 14).

We agree that from the modeling perspective, it can be interesting to further examine whether the difference between $[ML1p]$ in L1 and that in deeper layers is important for the *HAM* simulation in our model system. Thus, we have explored 50 additional sets of $[ML1p]$ and we have included the new simulation results in this revised manuscript (Supplementary Movie 15). Among them, we have tested 25 sets of $[ML1p]$ in a different range (L1: 1.05-1.5; Deep Layers: 0.6-2), including one set of $[ML1p]$ (L1:1.1; Deep Layers: 0.7). Further, we included 25 sets of $[ML1p]$ with no difference between L1 and deep layers in the range of 1.05-1.5, including one set of $[ML1p]$ (1.1 a.u. in all layers of SAM). All these 50 new sets of $[ML1p]$ lead to suppressed HAM mRNA expression in deep layers of SAMs, which are largely comparable to the simulation result using the initial set of $[ML1p]$ (1.1 a.u. in the L1 layer and 0.4 a.u. in deep layers), suggesting the suppression of HAM mRNA level when *ATML1* is ectopically activated in SAM is not sensitive to the difference between $[ML1p]$ in L1 and that in deep layers. We have included this conclusion in our current manuscript.

2. The authors should not call $D_{\{mir171\}}$ the diffusion constant. A diffusion constant has units $Length^2/Time$ and $D_{\{mir171\}}$ has the unit $1/Time$. Maybe one could call it rescaled diffusion constant (rescaled by the intrinsic length, probably cell size) or transport rate.

Response: Thanks for the comment. We have revised this term to rescaled diffusion constant (rescaled by cell size) in the current manuscript.

3. The symbol Δ should not be called Laplace operator, as this is used for the differential operator. Rather, one could call it discrete Laplace operator, or alike.

Response: Thanks for the comment. We have revised it to discrete Laplace operator in the current manuscript.

4. When the authors explain the quasi Monte Carlo sampling and the chosen intervals the talk about

arbitrary units, which I do not understand. I think all parameters have a unit, unless the authors non-dimensionalized them. In fact, most of the parameters used by the authors in the equations 1-5 have probably the dimension $1/h$.

Response: We thank the reviewer for helping us identify the imprecise description in our previous manuscript. Yes, all the parameter values that we randomly explore have their own units, which have been listed in Supplementary Table 1. We have revised the description in the manuscript to make them clear.